# The impact of Los Angeles basin pollution and stratospheric intrusions on the surrounding San Gabriel Mountains as seen by surface measurements, lidar, and numerical models

Fernando Chouza[1], Thierry Leblanc[1], Mark Brewer[1], Patrick Wang[1], Sabino Piazzolla[2], Gabriele Pfister[3], Rajesh Kumar[3], Carl Drews[3], Simone Tilmes[3], Louisa Emmons[3], and Matthew Johnson[4]

[1]Jet Propulsion Laboratory, California Institute of Technology, Wrightwood, CA, USA
[2]Jet Propulsion Laboratory, California Institute of Technology, 4800 Oak Grove Drive, Pasadena, CA, USA
[3]National Center for Atmospheric Research (NCAR), Boulder, CO, USA
[4]NASA Ames Research Center, Moffett Field, CA, USA

**Correspondence:** Fernando Chouza (keil@jpl.nasa.gov)

**Abstract.** In this work, the impact of Los Angeles basin pollution transport and stratospheric intrusions on the surface ozone levels observed in the San Gabriel Mountains is investigated based on a combination of surface and lidar measurements as well as WRF-Chem (Weather Research and Forecasting with Chemistry) and WACCM (Whole Atmosphere Community Climate Model) model runs. The number of days with observed surface ozone levels exceeding the National Ambient Air Quality

Standards exhibit a clear seasonal pattern, with a maximum during summer, when models suggest a minimum influence of stratospheric intrusions and the largest impact from Los Angeles basin pollution transport. Additionally, measured and modeled surface ozone and PM10 were analyzed as a function of season, time of the day and wind direction. Measurements and models are in good qualitative agreement, with maximum surface ozone observed for south-west and west winds. For the prevailing summer wind direction, slightly south of the ozone maximum and corresponding to south south-west winds, lower ozone

levels were observed. Back-trajectories suggest that this is associated with transport from the central Los Angeles basin, where titration limits the amount of surface ozone. A quantitative comparison of the lidar profiles with WRF-Chem and WACCM models revealed good agreement near the surface, with models showing an increasing positive bias as function of altitude, reaching 75% at 15 km above sea level. Finally, three selected case studies covering the different mechanisms affecting the near-surface ozone concentration over the San Gabriel mountains, namely stratospheric intrusions and pollution transport, are

analyzed based on surface and ozone lidar measurements, as well as co-located ceilometer measurements and models.

## 1 Introduction

A high concentration of near-surface ozone poses a hazard to human health (WHO, 2003), animals, and vegetation (Mauzerall and Wang, 2001). Although consistent efforts regulating the emissions of ozone precursors in the Los Angeles (LA) basin region have led to a considerable reduction in the near-surface ozone levels (Pollack et al., 2013), the LA basin is still marked

as a non-attainment area (EPA, 2020). For this reason, there has been an increased interest in understanding and modeling the

different processes driving the near-surface ozone concentration with the aim to generate more effective air quality regulation policies (Lin et al., 2017). Tropospheric ozone is mainly produced through photochemical processes involving volatile organic compounds in the presence of nitrogen oxides and sunlight (Monks et al., 2015). Additionally, stratospheric intrusions and elevated anthropogenic ozone plumes subject to long-range transport can also increase the troposphere ozone concentration

and can, in some specific conditions, affect the near-surface air quality (e.g. Lin et al., 2012b; Knowland et al., 2017; Langford et al., 2018).

The Los Angeles basin shows one of the highest near-surface ozone concentration records in the United States. This is a consequence of several combining factors, including high precursor emissions associated with transportation and industry, high temperatures, and abundant sunlight, as well as meteorological conditions and surrounding mountains that limit the venting of

the accumulated smog (Lu and Turco, 1996; Langford et al., 2010).

While somewhat limited by topography and meteorological conditions, long-range transport of LA basin pollution has been identified as a source of high ozone events around the Mojave desert and other locations further away (Langford et al., 2010; VanCuren, 2015). Among the processes driving the transport of LA basin pollution, we can find low-level transport through several passes found between the mountains that surround the LA basin as well as transport over these mountains and injection

in the free troposphere caused by the up-slope flow mechanism, also referred as the mountain chimney effect (Lu and Turco, 1996; Langford et al., 2010; De Wekker and Kossmann, 2015). While models and short-term measurements have been typically used to study these transport processes, no consistent long-term measurements have been conducted to quantify the frequency of these processes and investigate to which extent limited resolution models used for air quality forecasting reproduce them.

Additionally, many mountain-top monitoring stations have been typically assumed to sample free troposphere air and have

been used as part of a general effort to investigate long-term trends in background trace gas mixing ratios. While this assumption might be true in some stations or during particular periods, an assessment of the impact of local anthropogenic pollution sources is crucial to determine how well and during which periods this assumption of free troposphere sampling can be considered accurate (Lee et al., 2015; Tsamalis et al., 2014).

In this work, surface and lidar measurements conducted at the Jet Propulsion Laboratory Table Mountain Facility (JPL

TMF) in the San Gabriel Mountains (Southern California) are used to address three main objectives. Firstly, to demonstrate the new near-range measurement capabilities of the Table Mountain tropospheric ozone lidar and their value for pollution transport and deep stratospheric intrusion studies. Secondly, to investigate the relative impact of regional pollution transport and stratospheric intrusions on the exceedances of the National Ambient Air Quality Standards at TMF, and to determine the representativeness of surface measurements as a proxy for the free troposphere. Finally, to use these surface and vertical

profiles to evaluate the performance in complex terrain of the Weather Research and Forecasting model coupled with Chemistry (WRF-Chem) and Whole Atmosphere Community Climate Model (WACCM) forecasts produced daily by the Atmospheric Chemistry Observations and Modeling (ACOM) laboratory at the National Center for Atmospheric Research Atmospheric (NCAR).

The paper is organized as follows. Section 2 provides a general overview of the main characteristics of the LA basin region

and the datasets used in this paper, including a description of the ozone lidar and ceilometer, surface instruments deployed at

**Table 1.** Datasets used in this work together with the period under study, temporal resolution and variables analyzed. Except for the two models, all the instruments are located at JPL TMF.

| Dataset | Period | Temporal resolution | Variables used in this study |
|---|---|---|---|
| TMTOL | May 2019 - September 2020 | 1-hour averaging | $O_3$ profile (0.1 to ~15 km a.g.l) |
| CL51 | May 2019 - September 2020 | ~15 seconds | Attenuated backscatter and boundary layer height |
| Thermo Fisher 49i | Juanuary 2012 - September 2020 | 1 minute | Surface $O_3$ |
| Met One Model 212 | May 2019 - September 2020 | 1 minute | Particle counts (0.3 μm-10 μm) |
| TMF Met station | May 2019 - September 2020 | 1 minute | Surface temperature, humidity, pressure and wind |
| ACOM WRF-Chem | May 2019 - September 2020 | 1 hour | $O_3$, anthropogenic CO, boundary layer height and wind |
| ACOM WACCM | May 2019 - September 2020 | 6 hours | Stratospheric ozone |

JPL TMF, and the main setup characteristics of the WRF-Chem and WACCM models provided by NCAR ACOM. Section 3 presents an analysis of the relative impacts of pollution transport and stratospheric intrusions to the observed ozone threshold exceedances at TMF, as well as an evaluation of the WRF-Chem forecast based on ground and vertical profile measurements conducted between May 2019 and September 2020 (the period during which the model data is available). In section 4, three case studies depicting the main mechanisms driving high surface ozone events at JPL TMF are discussed and compared with the WRF-Chem/WACCM forecast of these events. Finally, a summary of the key findings of this paper is presented in Section 5.

## 2 Datasets and methods

### 2.1 Site description and data coverage

JPL Table Mountain Facility (34.38° N; 117.68° W, 2285 m a.s.l.) is located in the San Gabriel mountains (Fig. 1), north of the LA basin and 6 km northwest of Wrightwood, the closest town. The site hosts numerous instruments for air composition monitoring, including lidars and surface instruments (Table 1). Despite the high elevation, differential Optical Absorption Spectroscopy (DOAS) measurements by Chen et al. (2011) reported several days with signatures of anthropogenic pollution between late spring and early summer. Although these measurements already provide evidence of the LA basin impact at TMF, measurements based on DOAS have a limited capability to resolve the vertical extent of these anthropogenic layers. Similar conclusions have been reached by Gorham et al. (2010) at Mt. Wilson (34.22° N; 118.06° W, 1742 m a.s.l.), another high elevation site located in the San Gabriel mountains. In that case, ground-based measurements of CO and non-methane hydrocarbons showed repeated signatures of LA basin pollution transport, with a peak occurrence during summer months.

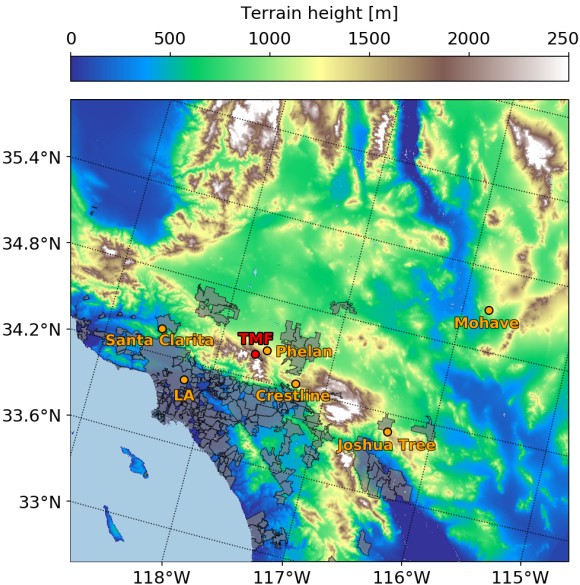

**Figure 1.** Terrain elevation of the LA basin and Mojave desert area. Urban regions are marked in grey. The location of TMF is marked in red, other surface ozone monitoring stations relevant for this study are marked in orange.

## 2.2 Table Mountain tropospheric ozone lidar (TMTOL)

TMTOL is an ozone differential absorption lidar (DIAL) that has been under operation at JPL TMF since 1991. Under its original configuration, the system was able to alternate between ozone and aerosol measurements (McDermid et al., 1991). In 1999, the system was redesigned to provide routine measurements of tropospheric ozone in the middle and upper troposphere for the Network for the Detection of Atmospheric Composition Change (NDACC) (McDermid et al., 2002), and later included in the Tropospheric Ozone Lidar Network (TOLNet). This modification included the removal of the aerosol measurement

capabilities, a larger telescope, and a receiver based on interference filters in place of the previous spectrometer-based one.

In 2018, TMTOL was fully automated and a new channel covering the range between 100 m a.g.l and 1000 m a.g.l was added to the system (Chouza et al., 2019). These two modifications greatly improved the capabilities of the system for air quality and transport studies in the PBL. Long runs of multiple hours or days can be routinely performed without over-stressing the operators, and the retrieved profile extends downwards into the PBL, in contrast to the former setup that allowed only

measurements down to 1000 m a.g.l.. In addition to the validation with tethered balloons reported in Chouza et al. (2019), routine validations of this new very-near-range receiver have been conducted with an UAV-borne electrochemical concentration cell (ECC) ozonesonde that provides ozone profiles between the ground and 120 m a.g.l.. The overlap of 20 m between the UAV measurements and TMTOL allows us to verify that the lidar measurement is not biased due to changes in the receiver overlap function. The results of these tests (36 in total) are summarized in Fig. 2. As can be seen, the difference between the

first valid TMTOL retrieval and the UAV measurements has been generally under 10%, which is in agreement with previous

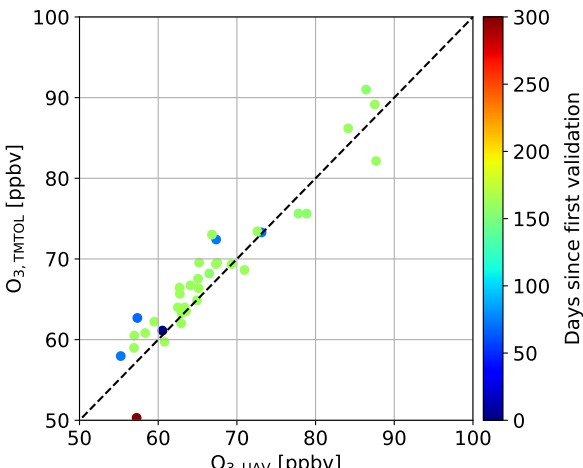

**Figure 2.** Comparison between the UAV-borne ozonesonde measurements and the corresponding TMTOL retrieval at 100-120 m a.g.l.. The scatter plot color indicates the number of days since the first UAV validation experiment (1 May 2019).

TMTOL validation studies (Leblanc et al., 2018) and indicate good stability of the very-near-range receiver performance over time.

## 2.3 Vaisala CL51 ceilometer

Among the atmospheric remote sensing instruments at TMF, a Vaisala CL51 has been operated almost continuously since 2015. This instrument, located approximately 30 m west of TMTOL, provides valuable co-located qualitative information regarding the near-surface aerosol layers as well as planetary boundary layer (PBL) height measurements (Wiegner et al., 2014). In this study, the PBL height derivation is obtained from the proprietary Vaisala algorithms included in the original ceilometer software. While the details of this algorithms are not available to the users, a general description can be found in Münkel and Roininen (2010).

## 2.4 Surface measurements

In addition to TMTOL and the ceilometer, a set of in-situ measurement instruments that provide near-surface measurements of ozone, particulate matter, and meteorological variables are currently deployed at TMF. In the case of surface ozone, measurements are collected by a Thermo Scientific Model 49i Ozone Analyzer that has been operated at TMF since 2013 with a brief interruption in 2016 due to problems with the instrument. The surface ozone photometer inlet is located at about 2 m a.g.l. on the north side wall of TMTOL building. Particulate matter measurements have been carried out since 2015 by a Met One Model 212, covering particle size from 0.3 μm to 10 μm. This instrument is located about 60 m south of TMTOL at an altitude of about 2 m a.g.l.. The PM10 values reported in this work have been obtained from the particle counter following Brattich

et al. (2020). Since the cut-off diameter of the particle counter is 0.3 μm, an underestimation in the derived PM10 values is expected for fine-mode dominated aerosol events. Finally, meteorological variables including temperature, humidity, and wind speed have been collected with the current setup since 2005 with only a few short interruptions driven by failures in the data acquisition system. The meteorological mast where the instruments are attached is approximately 30 m west of TMTOL. The wind speed and direction sensor is located at 10 m a.g.l., while the rest of the sensors are at 2 m a.g.l..

## 2.5 ACOM WRF-Chem forecast

The WRF-Chem air quality predictions are produced daily at NCAR using version 3.9.1 of the WRF-Chem model (Fast et al., 2006; Grell et al., 2005; Powers et al., 2017). The model domain is defined on a lambert conformal project with a horizontal grid spacing of 12 x 12 $km^2$. The model domain covers the contiguous United States (CONUS) with 390 and 230 grids points in longitudinal and latitudinal directions, respectively. The vertical grid in the model is composed of 43 levels stretching from the surface to 50 hPa. A detailed description (chemical and physical parameterization, emissions, driving meteorological and chemical fields) of the forecasting system configuration can be found at https://www.acom.ucar.edu/firex-aq/tracers.shtml and only details relevant to this study are summarized here.

Tropospheric ozone photochemistry is represented using the Model for Ozone and Related Tracers-4 (MOZART-4) chemical mechanism (Emmons et al., 2010). MOZART-4 contains 83 species that participate in 157 gas-phase reactions and 38 photolysis reactions. The model does not include stratospheric chemistry and lateral boundary conditions control the background as well as upper atmospheric concentrations. In addition, six carbon monoxide (CO) source tracers are included in the model to keep track of CO emitted from anthropogenic and biomass burning emission sources located inside the domain, photochemical production of CO from non-methane volatile organic compounds (NMVOCs) emitted within the domain, and background CO flowing into the domain produced by all non-CONUS sources including non-CONUS fires. CO tracers are subjected to the same physical and chemical losses (reaction with OH and deposition) as the standard CO species is but do not affect any atmospheric processes in the model.

The U.S. EPA National Emissions Inventory (NEI) 2014 is used to represent monthly varying anthropogenic emissions of trace gases and aerosols. No adjustments were made to the emissions due to COVID-19 related restrictions. Fire Inventory from NCAR (FINN) version 1 (Wiedinmyer et al., 2011) provides near-real-time (NRT) biomass burning emissions to the model, which are distributed vertically online within the model using a plume rise parameterization (Freitas et al., 2007). NRT FINN emissions are available with a latency of 1 day and are assumed to persist over the forecast cycle. The meteorological initial and boundary conditions are based on the 00 UTC cycle of the Global Forecast System (GFS) produced daily by the National Oceanic and Atmospheric Administration (NOAA). The chemical boundary conditions are based on the WACCM forecasts produced daily by NCAR (see Sec. 2.6 for WACCM details). The initial conditions for chemical fields are based on previous day's forecast. Hourly model output is saved for analysis. A two-day model forecast starts at 2 am MT every day and finishes in about 2 hours. Selected model output including concentrations of ozone, PM2.5, key precursor species, meteorological variables, and NRT observations of surface ozone and PM2.5 are displayed at https://www.acom.ucar.edu/firex-aq/forecast.shtml for dissemination to the public.

## 2.6 ACOM WACCM

The Whole Atmosphere Community Climate Model Version 6 (WACCM6) is one of the atmospheric components of the Community Earth System Model (CESM2) (Gettelman et al., 2019). WACCM6 is a fully coupled global Earth System model that extends from the Earth's surface towards the lower thermosphere ($\sim 150$ km altitude). The chemistry scheme is the MOZART Troposphere Stratosphere Mesosphere and Low Thermosphere Version 1 (TSMLT1) chemical mechanism (Emmons et al., 2020). The aerosol scheme is the Modal Aerosol Model (MAM4), including a volatility basis set (VBS) description of secondary organic aerosols (Tilmes et al., 2019).

For this study, it uses a horizontal resolution of 0.9° latitude x 1.2° longitude. The specified dynamics version used here adopts the levels of GEOS5 below 50 km and has a total of 88 vertical levels reaching to the model top. The simulation used in this study uses observed sea-surface temperatures and sea-ice conditions for present day that is coupled to the community land model Version 5 (CLM5). The atmospheric winds, temperature, and surface fluxes are nudged below 50 km towards NASA GMAO GEOS5.12 meteorological analysis with a Newtonian relaxation of 50 hours. Daily fire emissions are based on FINNv1. Anthropogenic emissions are from the CAMS Version 3 (Copernicus Atmosphere Monitoring Service) inventory. Biogenic emissions are derived using the Model of Emissions of Gases and Aerosols from Nature version 2.1 incorporated in the CLM (Guenther et al., 2012). A stratospheric ozone tracer is included in this configuration, which is set equal to ozone in the stratosphere and destroyed in the troposphere at the same rate as the model ozone (photochemical destruction and dry deposition).

This model version has been also used to provide a daily 10-day forecast since 2018, using GEOS5 meteorological forecast fields plots and output files that are available at https://www2.acom.ucar.edu/acresp/forecasts-and-near-real-time-nrt-products.

## 3 General ozone features and model evaluation

### 3.1 The impact of pollution transport and stratospheric intrusions on high ozone days

While Granados-Muñoz and Leblanc (2016) already provided an overview of the surface ozone characteristics at TMF for the period 2013-2015, additional analysis as well as supporting model information are expected to help to better characterize the impact of the LA basin pollution at TMF. Figure 3a shows a histogram of ozone exceedances based on the surface ozone measurements conducted at TMF from January 2012 to September 2020. Due to a malfunction of the TMF surface ozone analyzer, measurements from 2016 are excluded from this study. While variable from year to year, the number of days with ozone levels exceeding the EPA regulations follow a clear progression over the months, with almost no exceedances during winter and a large number of exceedances during summer. A remarkably low number of exceedances were observed in 2019, which is likely associated with below-average temperatures (not shown). Since the local production of ozone precursors is very limited in the San Gabriel mountains, the origin of these exceedances is likely related to direct transport from the LA basin region as previously reported in the case study presented in Langford et al. (2010). Together with anthropogenic pollution transport, stratospheric intrusions have also been pointed as the cause of high surface ozone events in the region (Lin et al.,

2012a). Since no isotope-based stratospheric tracer measurements (Stohl et al., 2000) are available at TMF, the surface impact of stratospheric intrusions is hard to quantify. Nevertheless, Fig. 3b provides an overview of the stratospheric tracer reported by WACCM at 2700 m a.g.l. since May 2019 over TMF. This figure shows a clear pattern in the number of deep intrusions, with a maximum during winter and early spring, and a minimum during summer and early fall. This seasonality is in agreement with previous studies in the region (Granados-Muñoz and Leblanc, 2016) and northern hemisphere (Sprenger and Wernli, 2003). On the other hand, Fig. 3c provides an overview of the surface anthropogenic CO contribution forcasted by WRF-Chem, which shows larger median values during summer, when transport of LA basin pollution by upslope flow is expected to be at its maximum. Together with these box plots, the values of the stratospheric ozone contribution reported by WACCM and WRF-Chem anthropogenic CO during exceedance days are also shown. In most cases, the WACCM stratospheric ozone levels during exceedance days fall within 1.5 times the inter-quartile range (IQR), which suggests that it plays a limited role in most ozone exceedance cases. On the other hand, anthropogenic CO during ozone exceedance days is typically well above the median, which suggests that pollution transport plays a more important role in the ozone exceedances than the stratospheric intrusions. Similarly, relative humidity and temperature are also presented, as intrusions are typically associated with cold fronts and dry air. Temperature and humidity values during ozone exceedance days do not show a major departure from median values. While this analysis does not rule out stratospheric intrusions as the cause of some of the exceedances, their overall impact appears to be limited according to the WACCM and WRF-Chem models. Additionally, the concentrations of the anthropogenic CO tracer as forecast by WRF-Chem appear to be highly variable and generally non-negligible over the whole year, with median values of 12 ppbv during winter and over 25 ppbv during summer, suggesting that surface and near-ground measurements at TMF are strongly influenced by local sources and cannot be generally assumed to be representative of the free troposphere.

## 3.2 Surface ozone and PM10 as function of time and wind direction

Since LA basin pollution transport seems the most likely cause for the large number of exceedances observed during the March-October period, an analysis of the ozone and PM10 levels as a function of local time and wind direction is presented in Fig. 4 for the period between summer 2019 and summer 2020. Here, summer, fall, winter and spring are defined as June-July-August (JJA), September-October-November (SON), December-January-February (DJF), and March-April-May (MAM) respectively. As supporting information, ozone, PM10, and anthropogenic CO provided by the ACOM WRF-Chem forecast is also included for the same period. The first and fourth rows, which present the number of values observed and forecast for each time of the day and wind direction, reveal prevailing winds almost exclusively from the south south-west (SSW) during summer and spring with very little temporal variability. During fall and winter, a second prevailing wind direction can be seen coming from the east north-east (ENE). WRF-Chem shows a very similar pattern, with fewer SSW points during winter and slight changes in the prevailing wind directions.

In the case of the ozone measurements and forecast (second and fifth rows), the largest mean values appear in the summer early night extending in some cases into the next day. In the case of WRF-Chem, the peak in the mean ozone has a similar amplitude but is approximately 3 hours earlier than observed and minimum ozone values during morning time are significantly lower (about 10 ppbv) than the measured ones, which leads to an overestimation of the ozone diurnal cycle amplitude by the

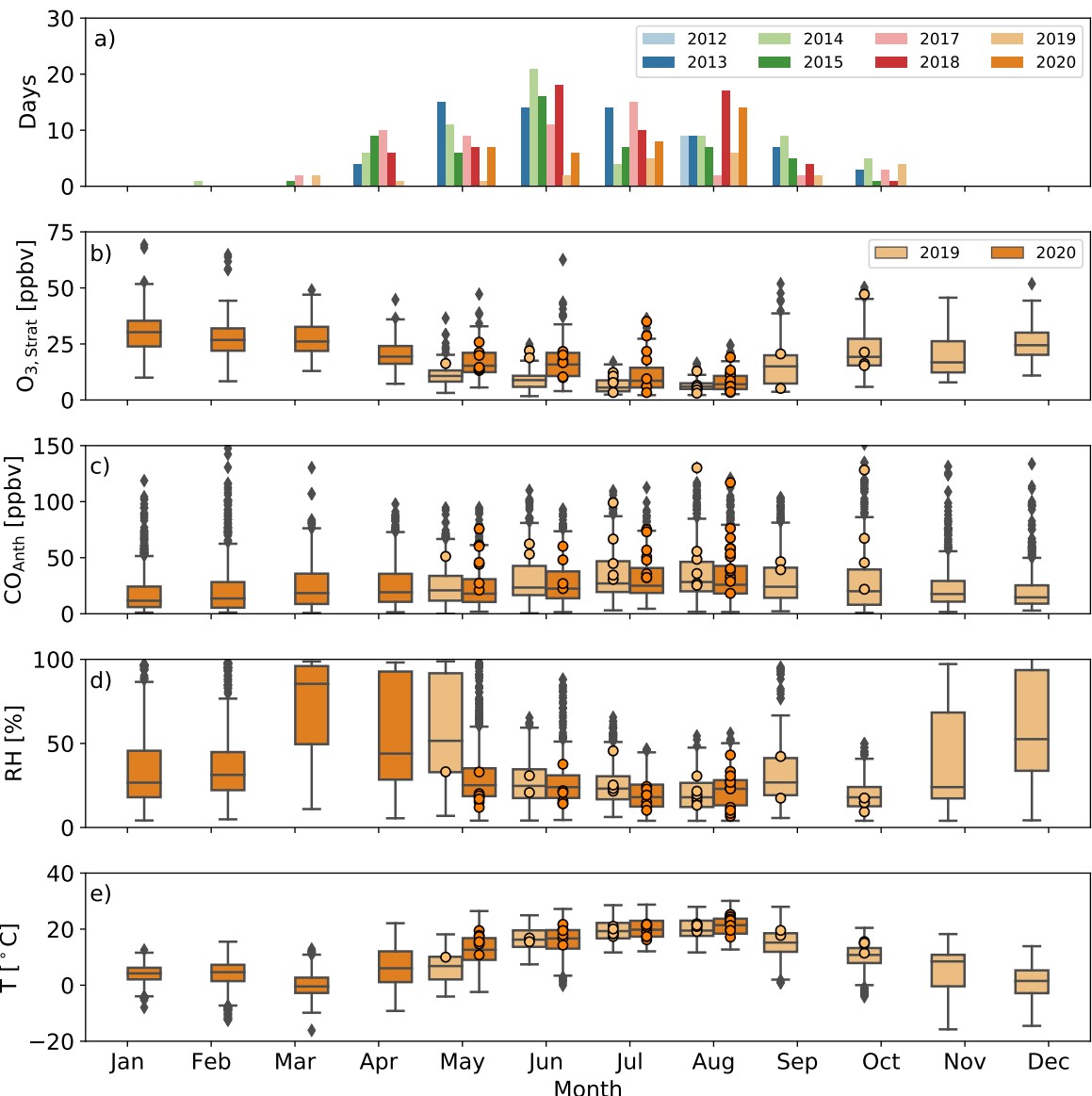

**Figure 3.** (a) Number of days exceeding the EPA maximum ozone regulations (>70 ppbv 8h MDA) as function of month and year between 2012 and June 2020 derived from the TMF 49i surface ozone monitor. (b) WACCM forecast of the stratospheric ozone at 2700 m a.s.l. as function of month and year between May 2019 and June 2020 presented as a box plot. (c) WRF-Chem anthropogenic CO levels for the same period shown in (b). (d,e) Measured surface relative humidity and temperature for the same period presented in (b). WACCM stratospheric ozone, WRF-Chem anthropogenic CO, relative humidity and temperature during the exceedance days are shown for comparison (light and dark orange dots). Values exceeding 1.5 times the interquartile range (whiskers) are also shown (black diamonds).

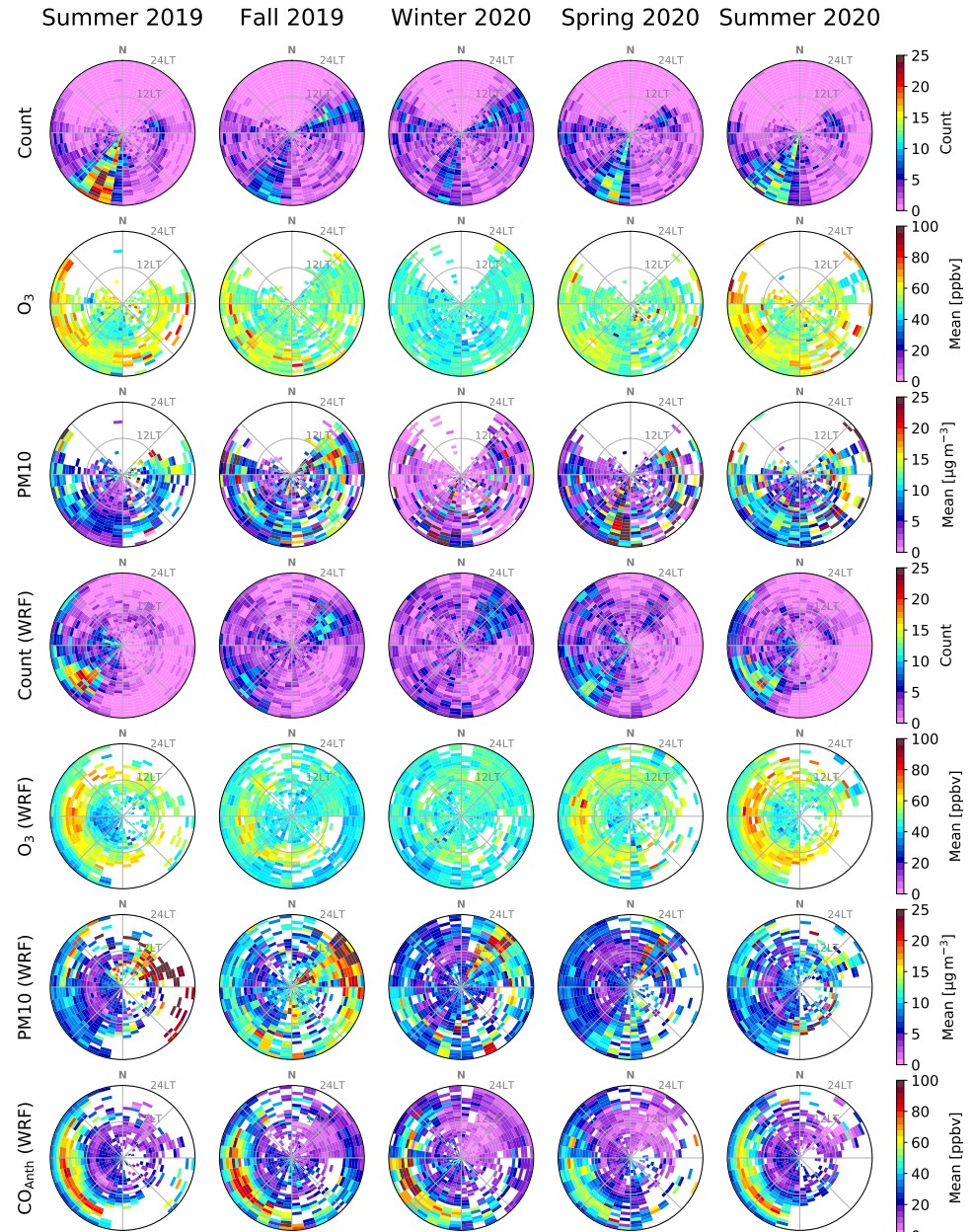

**Figure 4.** Overview of the number of surface measurements (first row), mean ozone (second row) and mean PM10 (third row) at TMF as function of the time of the day (radial direction, local time), wind direction (angular) and season (columns) for the period comprehended between June 2019 and September 2020. The corresponding values forecast by WRF-Chem are presented in the fourth, fifth and sixth rows, with the addition of the anthropogenic CO contribution (last row).

model. A comparison of the WRF-Chem surface ozone output with the nearby surface ozone stations of Phelan and Crestline, suggests that this temporal shift in the ozone maximum is a particular feature of the model over TMF, while the underestimation

of the ozone levels during morning hours by WRF-Chem is common to all three stations (Fig. A1). The cause of this localized temporal shift is uncertain at this point, but it might be related to the smoothing of the terrain in the model and its impact on the upslope flow.

The ozone mean is observed to peak for south-west and west winds, slightly north from the prevailing wind direction. For the prevailing wind direction, the observed ozone levels during summer 2019 and 2020 show surprisingly low values, about 10

215 ppbv below the values observed for west winds. In the case of the forecast, this feature is well reproduced for summer 2019, while for summer 2020, this feature is less pronounced (note that neither WACCM nor WRF-Chem have emissions adjusted for COVID-related restrictions). To understand the origin of this difference, HYSPLIT/PySplit (Stein et al., 2016; Warner, 2018) back-trajectories based on WRF-Chem meteorological fields for summer 2019 (Fig. 5) and summer 2020 (Fig. 6) were calculated starting at 2:00 UTC (peak of surface ozone at TMF) for 4 hours, which corresponds to the typical time difference

with respect to the ozone maximum in the central LA area (Granados-Muñoz and Leblanc, 2016). These trajectories were separated into two groups, one corresponds to the days where the wind direction was between 250 and 300 degrees (Figs. 5 and 6, first column) and a second group, where the wind direction was between 210 and 240 degrees (Figs. 5 and 6, second column). For both groups, the trajectories were started at 10 m above TMF. Additionally, the mean modeled surface ozone (Figs. 5a,b and 6a,b), $NO_2$ (Figs. 5c,d and 6c,d) and PM10 (Figs. 5e,f and 6e,f) at the time of the end of the back-trajectories (4 hours

before the start, peak ozone in central LA) were calculated for these two groups. The surface ozone pattern, as reported in Lu and Turco (1996), is similar for these two cases, with high ozone in the Santa Clarita/San Fernando Valley area and the eastern LA basin and relatively low surface ozone in the central LA area. The comparison with the EPA surface ozone monitoring stations (see LA, Santa Clarita, Crestline, and Phelan measurements in Figs. 5a,b and 6a,b) shows a good qualitative agreement with the WRF-Chem output, but with measurements showing generally higher ozone levels at Crestline/Santa Clarita and lower

values at the central LA site. The latter can be attributed to enhanced near-surface titration associated with high surface $NO_x$ levels (Figs. 5c,d and 6c,d). The trajectories corresponding to the prevailing winds (210-240 degrees) end over this high surface $NO_2$ region, which likely explains the difference with respect to the 250-300 degrees back-trajectories that mainly end in the San Fernando Valley area. During fall and spring similar patterns are observed, but with an overall lower ozone concentration. As in the summer case, the forecast nighttime ozone concentration is lower than the observed one. During winter, forecast and

observed ozone exhibits the largest homogeneity among seasons, with very little dependence on time or wind direction.

During summer 2019, PM10 observations and forecast (third and sixth rows of Fig. 4) exhibited a pattern very similar to the one observed for ozone, with a diurnal cycle peaking in the afternoon for west and north-west winds. For the prevailing SSW winds, a minimum in the PM10 is observed and forecast. A difference is observed for ENE winds, where a second PM10 maximum in the afternoon can be distinguished, likely associated with dust transported from the Mojave desert. During fall

and spring, and although there is still a qualitative agreement between measurements and forecast, the forecast PM10 values are generally larger than the measured ones. The winter PM10 measurements show very little aerosol load, while WRF-Chem shows a pattern similar to spring and summer. In contrast to the observations and forecast presented for summer 2019 and the

forecast for summer 2020, summer 2020 PM10 observations shows generally larger concentrations and no minimum associated with south south-west (SSW) winds. The explanation for this discrepancy is unknown, but might be associated with enhanced aerosol load product of extensive wildfires occurred during summer 2020.

Finally, the anthropogenic CO forecast (last row) provides a clear view of the LA transport process, with relatively low seasonal variability (as discussed on Sec. 3.1), a clear maximum in the late afternoon and south to south-west winds.

WRF-Chem was shown to be able to qualitatively reproduce most of the features observed in the spatio-temporal distributions of ozone and PM10 at TMF and the surrounding stations. Some differences were observed regarding the amplitude of the ozone diurnal cycle at TMF and the nearby stations (Figs. 4 and S1). In the particular case of TMF, the forecasted maximum of the ozone diurnal cycle was about 3 hours earlier than the measured maximum. As also shown in Sec. 3.1, the results shown in this subsection indicate that surface ozone concentrations during summer at TMF are strongly influenced by pollution transport from the LA basin region. Transport from central LA, where titration limits the surface ozone concentrations, is characterized by generally lower ozone levels at TMF, while transport from Santa Clarita is generally associated with higher ozone concentrations.

### 3.3 Vertical ozone profile

While surface measurement instruments like the 49i Ozone Analyzer and the particle counter provide almost continuous datasets (Sec. 3.2), vertical profiles, as obtained by lidars and balloons, are often required to understand the vertical extent, source, and potential for long-range transport of different types of high surface ozone events. In the case of TMF, this is especially true, as the surrounding mountainous terrain adds a layer of complexity to the transport processes and the interpretation of surface measurements.

The previous long-term study presented in Granados-Muñoz and Leblanc (2016) focused on the free troposphere, as the minimum TMTOL range was mainly limited to 1.3 km a.g.l., which left out of the study most of the PBL. The new very-near range channels, able to reach as low as 100 m a.g.l., have been operated and validated in a routine manner since their installation in mid-2018, allowing almost complete coverage of the PBL. Additionally, during the last two years, longer run periods of multiple days have been regularly conducted by TMTOL in order to capture forecast SI and LA pollution transport events with the aim to understand the relative contribution of these to the observed exceedances at TMF and investigate to what extent the forecasting tools are able to reproduce them.

While validation of surface ozone and PM forecast is conducted as part of the ACOM WRF-Chem model runs on a routine basis, vertical evaluation is typically restricted to specific locations, as this type of measurement is not nearly as common as the surface ones. In addition to the multi-day runs focused on particular events, TMTOL has continued its regular operations consisting of one hour daily measurements during TROPOMI overpasses (typically around 1 pm local time) and 2-hour measurements 4 to 5 times per week during early nighttime. In this section, all the profiles retrieved from TMTOL since the beginning of the ACOM WRF-Chem forecast runs (May 2019) are used to evaluate the general performance of the model over TMF.

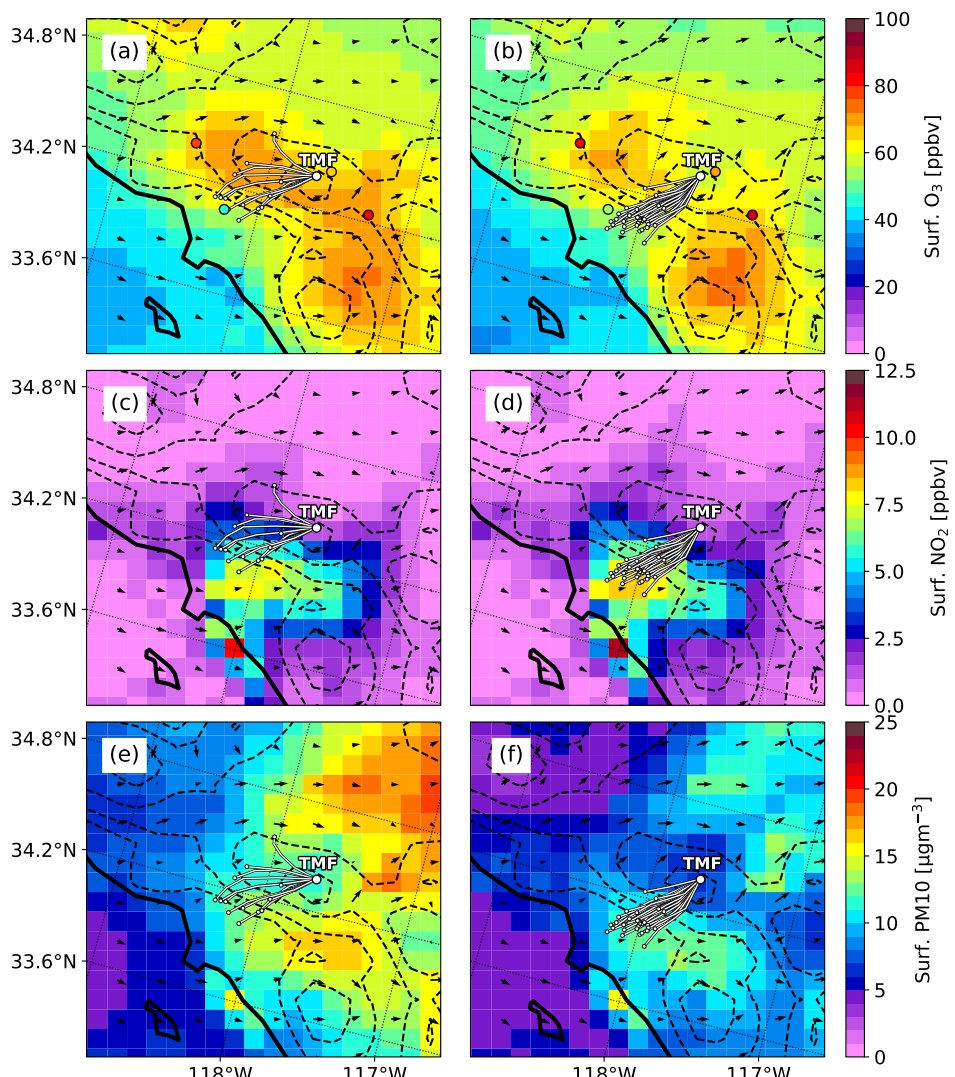

**Figure 5.** Back-trajectories (white with black borders) started at TMF at 2:00 UTC and ended at 22:00 UTC (-1 day) are shown together with the corresponding mean modeled surface $O_3$ (a,b), $NO_2$ (c,d) and PM10 (e,f) at the time of the end of the trajectories for two TMF wind direction groups during summer 2019. (a,c,e) Trajectories started on days where the surface winds over TMF were between 250 and 300 degrees at 2:00 UTC. (b,d,f) Trajectories started on days where surface winds over TMF were between 210 and 240 degrees. 10-m winds are also shown (black arrows). Elevation contours are shown (dashed black) for 250, 500, 1000 and 1500 m a.s.l.. The mean measured surface ozone values are shown as dots (black border) for LA, Santa Clarita, Phelan, and Crestline stations in panels (a,b).

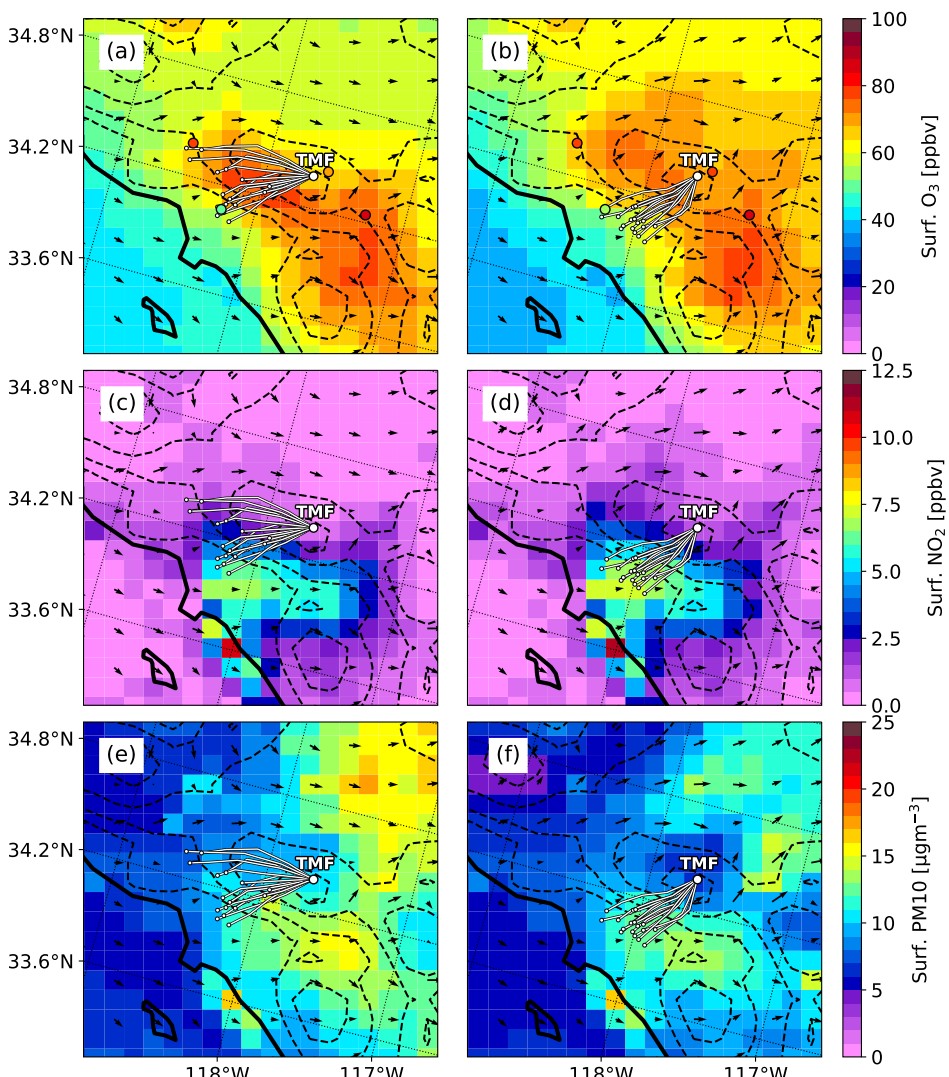

**Figure 6.** Same as Fig. 5, but for summer 2020.

Due to the complex nature of the terrain surrounding TMF, an assessment of the impact of the terrain smoothing associated with the limited spatial resolution of WRF-Chem has to be made. Figure 7 presents an overview of the WRF-Chem terrain elevation in the LA basin area, together with two cross-sections showing the difference between the actual and WRF-Chem terrain elevations. The A-A' cross-section is selected to be parallel to the prevailing winds in the region, while the B-B' is

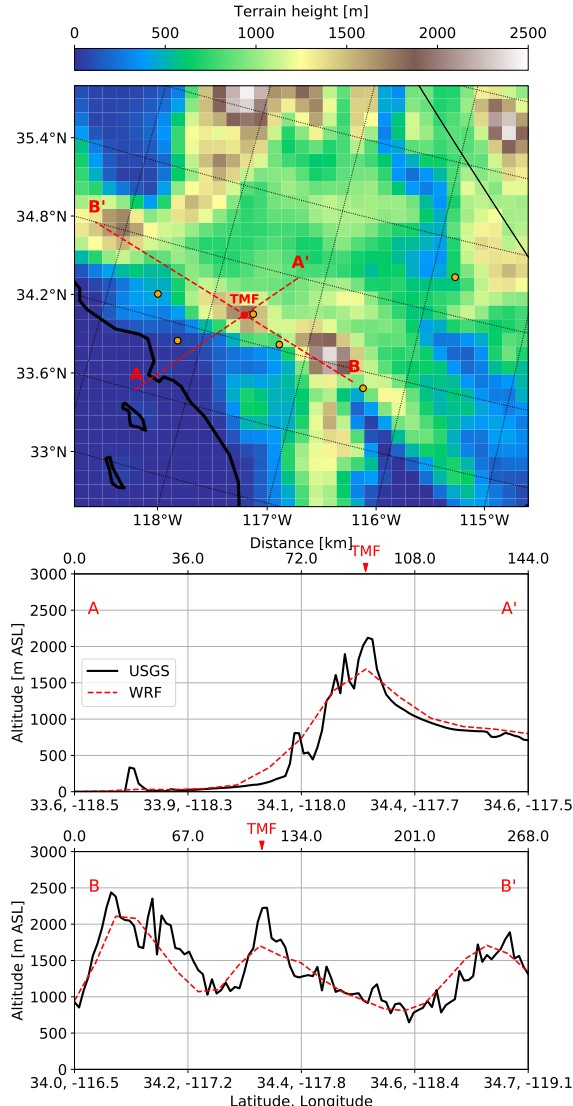

**Figure 7.** Overview of the terrain elevation used as part of the WRF-Chem simulation for the area under study. The TMF location, as well as the actual (solid, black) and WRF-Chem (dashed red) elevation profiles are shown two selected cross-sections used in this study.

almost perpendicular to A-A' and provides a general view of the passes and mountains that affect the outflow of the LA basin pollution. Both cross-sections were selected to run over TMF.

Thanks to the relative smoothness of the mountains surrounding the LA basin, a fairly good agreement between the actual terrain and the model terrain can be seen along the two cross-sections. The difference at TMF is among the largest ones, with the actual elevation being about 2.3 km and the modeled one about 1.7 km. Since this difference is mainly restricted to the top of the mountain, the effect on the mountain venting process and the associated vertical ascent is expected to be small.

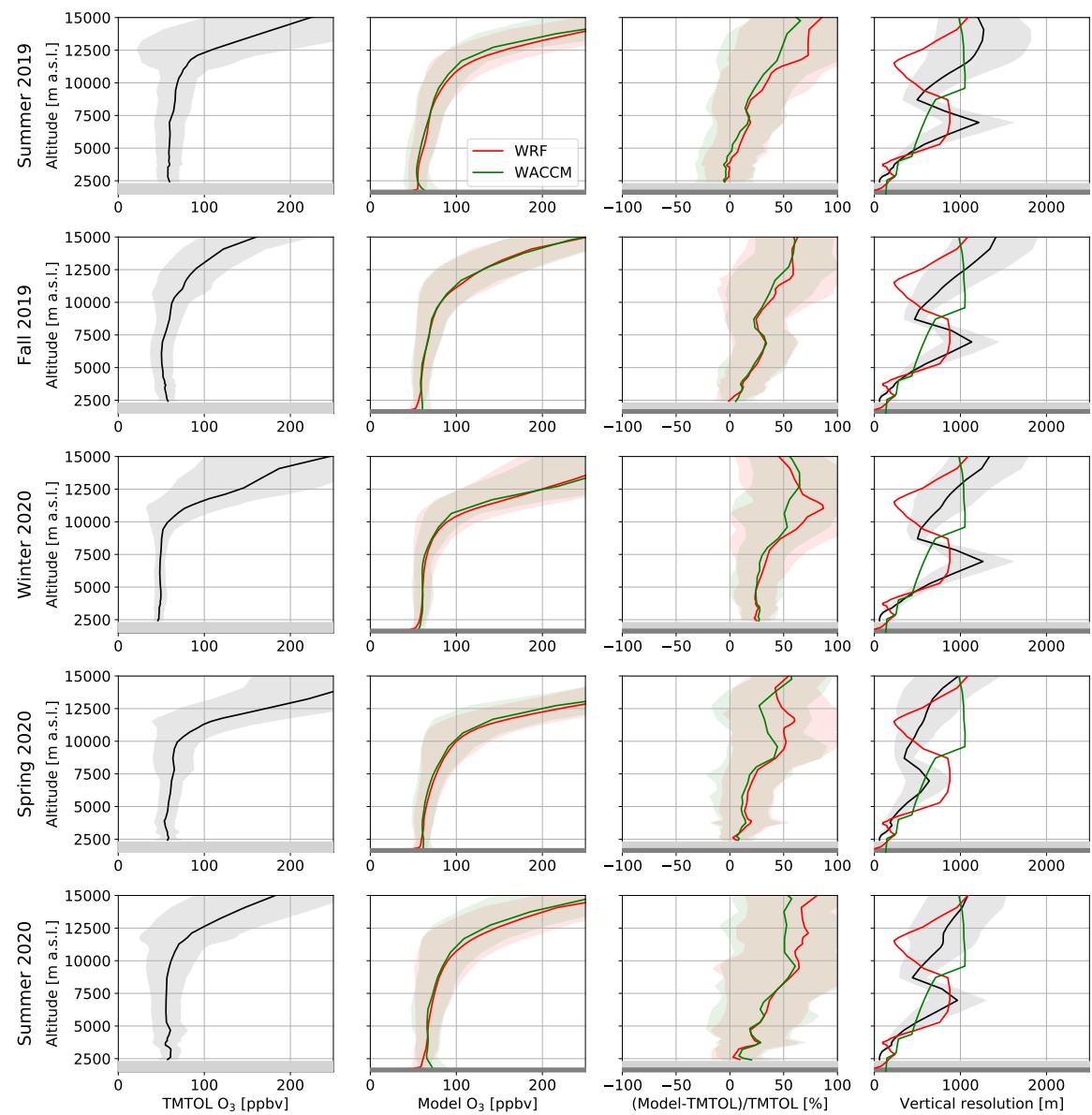

**Figure 8.** Relative difference (third column) between TMTOL (first column), WRF-Chem (red) and WACCM (green) ozone profiles (second column) for the period between summer 2019 and summer 2020 (rows). TMTOL retrieval vertical resolution as well as the vertical grid of the models are also shown for each season (fourth column). Actual ground level and WRF-Chem surface level are shown as grey and dark grey shaded areas respectively. 1-sigma variability on the ozone profiles and vertical resolution of TMTOL is indicated by the shaded areas.

Figure 8 presents an overview of the comparison between TMTOL, WRF-Chem and WACCM for the same period analyzed in Sec. 3.2 (summer 2019 to summer 2020), with each row of the plot corresponding to one season. The first column of Fig. 8

shows the mean and standard deviation (1-sigma) of the TMTOL profiles available for each season (a total of 726 profiles). The second column shows the mean of the temporally closest WRF-Chem and WACCM profiles (excluding profiles corresponding to the model spin-up period) to each TMTOL profile included in the mean shown in the first column, as well as the standard deviation of these profiles. The third column presents the mean of the relative difference between each individual TMTOL and corresponding WRF-Chem and WACCM profiles, as well a the standard deviation of these differences. In order to perform this comparison, the TMTOL profiles are averaged over each level of the models grids. As supporting information, the mean vertical resolution (and corresponding standard deviation) for the TMTOL retrieval (Leblanc et al., 2016), is presented together with the WRF-Chem and WACCM models vertical resolution in the fourth column. It is important to notice that the TMTOL vertical resolution is mainly controlled by the signal-to-noise ratio and the specified retrieval uncertainty. For a fixed uncertainty, the vertical resolution degrades as the amount of solar background increases. For this reason, the vertical resolution during daytime experiments is generally lower than during nighttime (see Fig. S2). Additionally, since TMTOL consists of different receivers looking at different altitude ranges, the vertical resolution changes as a function of altitude. This change in receivers can be seen as sharp changes in the vertical resolution at around 3 km and 7-8 km.

Overall, the WRF-Chem forecast shows an excess of ozone across the full range under analysis, independent of season and time of the day (not shown). This excess is limited to about 25% in the PBL, but increases almost monotonically with altitude reaching differences of up to 75% at 15 km. No clear difference in the model bias behavior was observed across the tropopause, typically found between 12 km a.s.l. (winter) and 16 km a.s.l. (summer). With respect to the seasonal trends, a good qualitative agreement between the model and TMTOL is observed, with an ozone increase in the UTLS (upper troposphere lower stratosphere) region, with lower and sharper transitions than during winter. The observed variability in the free troposphere also shows a clear seasonal dependence, with larger variability during summer and reduced variability during winter. The same seasonal pattern is also visible in the WRF-Chem profiles.

The WACCM forecast shows a very similar behavior to the one described for WRF-Chem, including the altitude-dependent bias. Since the WRF-Chem chemical boundary conditions are determined by the WACCM forecast (Sec. 2.5), the bias observed in the WRF-Chem runs is likely a result of the bias in the WACCM forecast. In order to investigate if this ozone excess is a particular feature of WACCM over TMF, we performed a comparison of the WACCM ozone forecasts and the ECC ozonesondes launched regularly at Trinidad Head, California (about 1000 km north-west from TMF) and Boulder, Colorado (about 1200 km north-east of TMF) for May 2019 to August 2020. The results (Fig. 9) indicate a similar altitude-increasing bias, suggesting a synoptic scale deviation of the forecast for the period under study as a possible reason.

## 4   High surface ozone drivers at TMF

The previous section presented a general overview of the ozone and surface PM10 characteristics at TMF as well as an evaluation of the WRF-Chem capabilities to reproduce them. In this section, three case studies are presented to illustrate specific mechanisms by which the surface or near surface ozone concentration at TMF can be affected, and investigate the extent to which WRF-Chem is able to reproduce them.

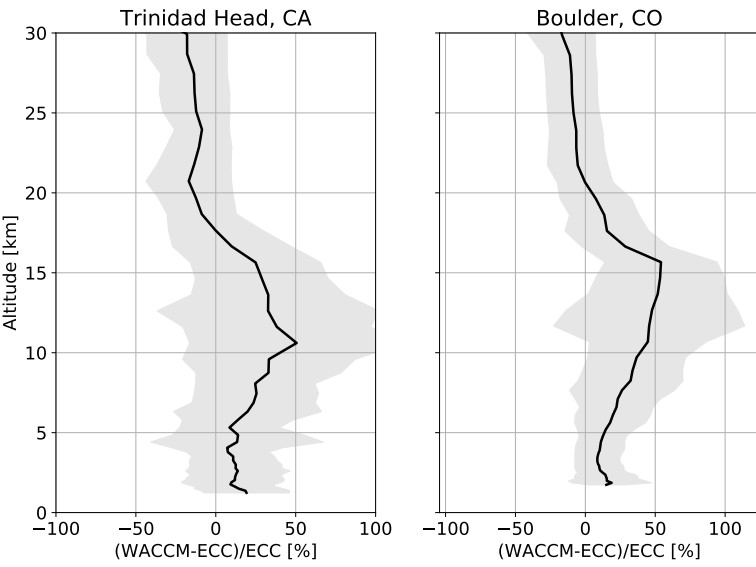

**Figure 9.** Relative mean difference between WACCM and ECC ozonesondes during the period comprehended between May 2019 and August 2020 over Boulder, Colorado and Trinidad Head, California. The 1-sigma standard deviation of the difference is indicated by the shaded area.

## 4.1 27-29 May 2020: A LA pollution transport event

Between 27 May and 28 May, two TMTOL extended runs were decided based on the WRF-Chem forecast of high surface ozone and anthropogenic CO levels (grey-shaded contours) associated with LA basin pollution transport (Fig. 10a). For the early part of 27 May (the late afternoon of 26 May if local time is considered), low levels of stratospheric influence were forecast by WACCM (hatched contours), while during the late part of 27 May and beginning of 28 May, a common case of LA basin pollution transport was expected with no stratospheric influence below 6 km a.s.l.. In this latter case, simulated surface ozone levels were compatible with a surface ozone exceedance event, which triggered the second extended TMTOL run. Figure 10b presents an overview of these two measurement sections, as well as the profile captured by an ozonesonde launched on 28 May 1:22 UTC (Fig. 11, solid) and the ceilometer-derived PBL height (Fig. 10c). During the first measurement section conducted between 27 May 1:24 UTC and 7:30 UTC, homogeneous ozone levels of about 70 ppbv were observed in the PBL, while the free troposphere was characterized homogeneous levels of about 40 ppbv. While TMTOL measurements and WRF-Chem are in qualitative agreement, the observed high ozone levels extended later in the day than the forecast ones, which is compatible with the general behavior presented in Sec. 3.2. The ceilometer backscatter measurements presented in Fig. 10c show a strong correlation with the TMTOL profiles, with high ozone in the PBL associated with high aerosol levels. Finally, the first part of this case study revealed a good agreement between the forecast and the surface measurement in the peak ozone mixing ratio (Fig. 10d).

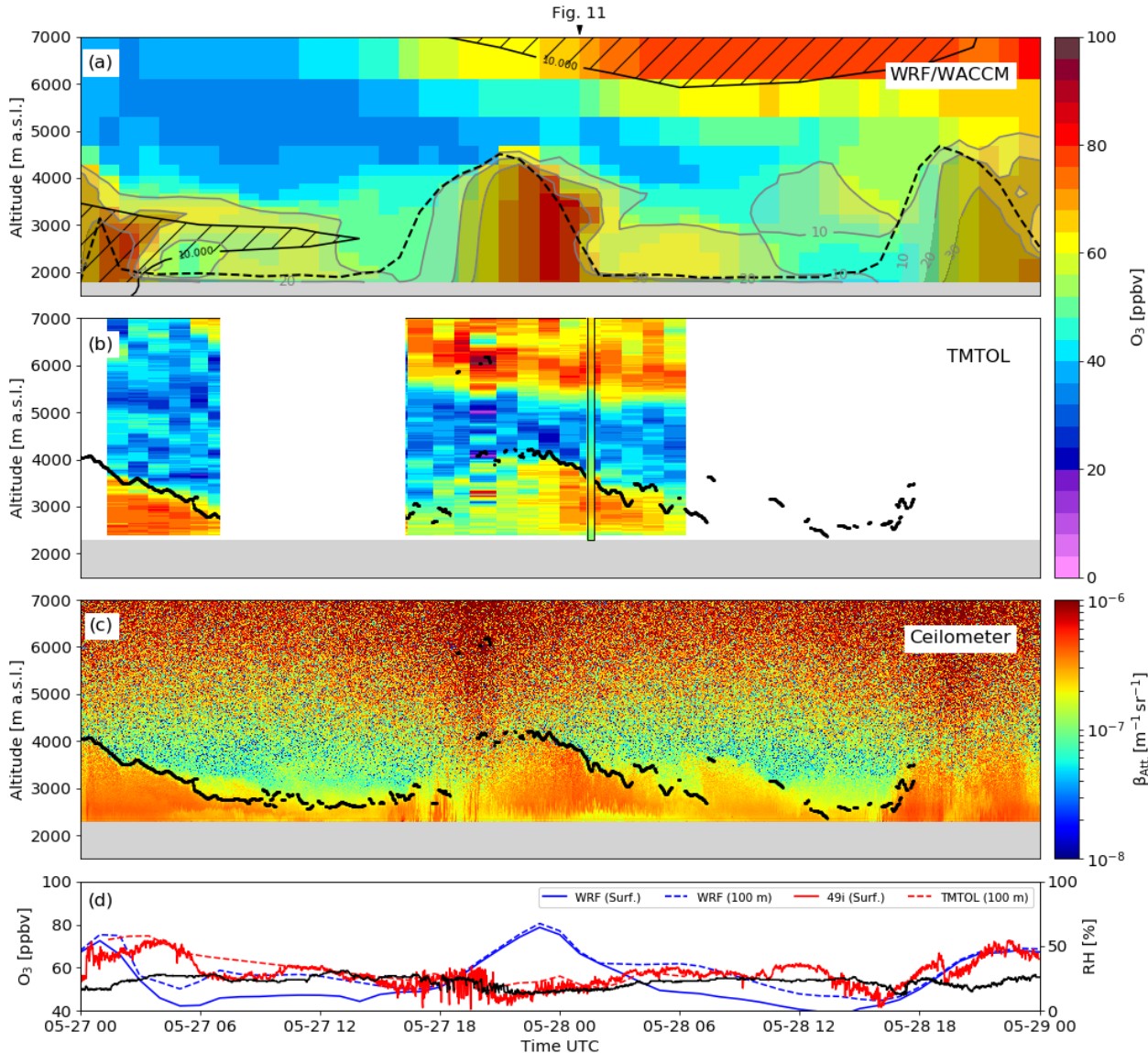

**Figure 10.** Overview of the model outputs and measurements over TMF between 27 May and 29 May UTC. (a) WRF-Chem ozone mixing ratio over TMF (color scale), WRF-Chem anthropogenic CO (grey shaded contours), PBL height (dashed black) and WACCM stratospheric tracer (hatched contours). (b) TMTOL measurements (color scale) and PBL height (black points). (c) Ceilometer-derived attenuated backscatter (color scale) and PBL height (black points). (d) WRF-Chem ozone mixing ratio at surface (solid blue) and 100 m a.g.l. (dashed blue) together with the 49i Ozone Analyzer surface ozone measurements (solid red), TMTOL ozone mixing ratio retrieval at 100 m a.g.l. (dashed red) and surface relative humidity (solid black). The ozonesonde profile is shown overlaid at the time of the launch and surrounded by a black box.

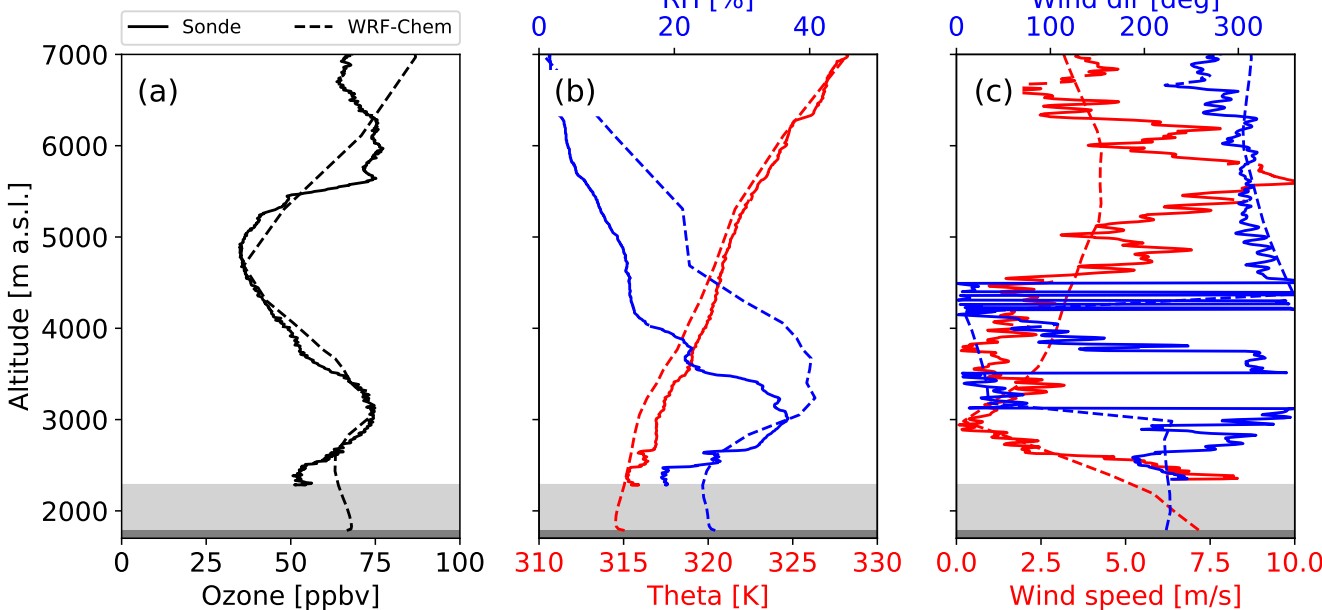

**Figure 11.** Sonde and corresponding WRF-Chem profiles for the launch conducted on 28 May 1:22 UTC. (a) Sonde ozone profile (solid black) and corresponding WRF-Chem output (dashed black). (b) Sonde-derived potential temperature (solid red) and relative humidity (solid blue) together with the corresponding WRF-Chem profiles (dashed, same colors). (c) Sonde-derived wind speed (solid red) and direction (solid blue) together with the corresponding WRF-Chem profiles (dashed, same colors). The actual TMF elevation (light grey shaded) is shown together with the model elevation (grey shaded).

In the case of the second TMTOL measurement period, conducted between 27 May 16:30 and 28 May 6:30 UTC, the PBL ozone mixing ratio follows a similar spatio-temporal progression as in the previous measurement section, including a similar temporal displacement in the ozone maximum with respect to the model forecast (see Sec. 3.2), similar PBL height,
and comparable ozone levels in the PBL and free troposphere. Despite these similarities, there is an interesting feature that makes this case specially interesting for air quality forecasting. While the simulated surface ozone was expected to exceed the 70 ppbv EPA standard at the transition between 27 May and 28 May, and high ozone levels were measured by TMTOL above 300 m a.g.l., almost no surface impact has been measured associated with the LA pollution transport event (Fig. 10d).

The origin of this discrepancy was traced back to two well-mixed near-ground layers bounded by very sharp temperature
inversions (Fig. 11b) that prevented the ozone-laden air from LA from being down-mixed and affecting thus the surface ozone concentration. The inversions that bounded these two layers were found at 2480 m a.s.l. and 2660 m a.s.l.. The first layer, the closest one to the ground and about 200 m deep, was characterized by an ozone mixing ratio of about 55 ppbv and relative humidity of 20 %, while the second layer, about 150 m deep exhibited an ozone mixing ratio of about 65 ppbv and relative humidity in the order of 27 %. Sitting on top of these two layers, we can see a 1.5 km deep layer characterized by relative high

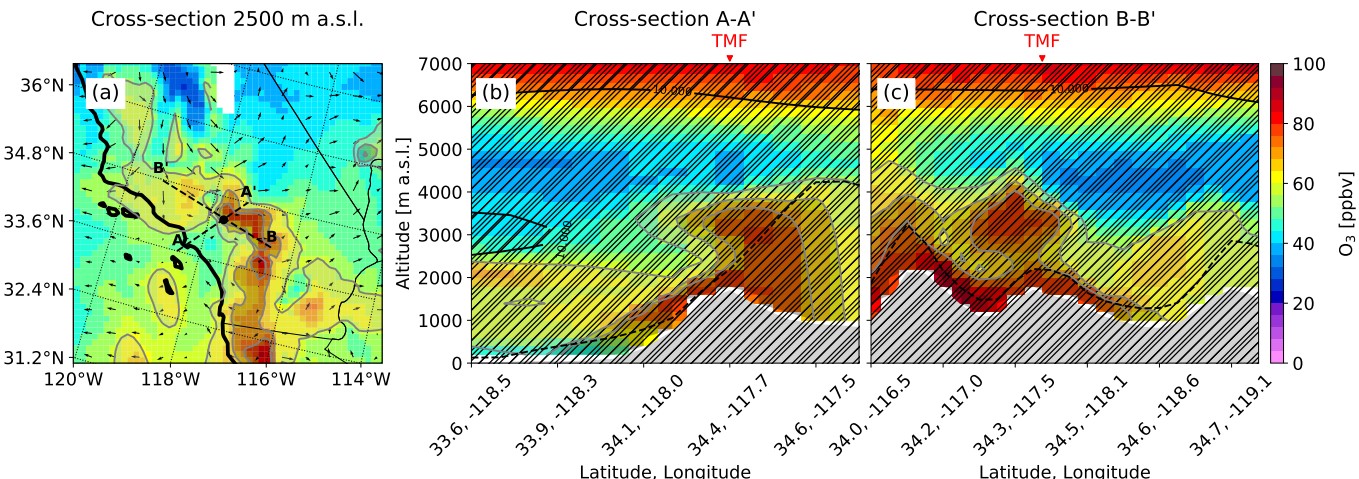

**Figure 12.** Horizontal and vertical cross-sections of WRF-Chem and WACCM forecast outputs for 28 May 1:00 UTC. (a) Horizontal cross-section at 2.5 km a.s.l. (b) Vertical cross-section along A-A' line. (c) Vertical cross-section along B-B' line. Total forecast WRF-Chem ozone concentration is shown in a color scale (all panels). WRF-Chem anthropogenic CO levels (grey-shaded contours, all panels), WACCM stratospheric ozone (hatched contours, (b) and (c)), PBL height (dashed black), and WRF-Chem winds (arrows, (a)) are also shown.

ozone mixing ratio (75 ppbv peak at 3 km a.s.l.) and a higher relative humidity that the other two layers (37 % at 3 km a.s.l.). The ceilometer profiles captured during the second TMTOL experiment shown in Fig. 10c exhibit a similar structure, with a relatively aerosol-poor layer below 2660 m a.s.l. and an aerosol-laden layer for the rest of the PBL. Finally, the wind profile presented in Fig. 11c revealed high north-westerly winds at low levels, which rapidly slow down and turn to northerly direction above 2660 m a.s.l.. While WRF-Chem forecast (Fig. 11, dashed) was able to forecast most of the previously mentioned

features captured by the ozonesonde, including the undercutting of the LA basin pollution layer by an air mass characterized by lower ozone mixing ratio and lower humidity, the terrain smoothing caused this layer to have a limited impact on the surface, which translated into a large difference with regard to the forecast surface ozone.

   In order to provide a broader context to this case study, Fig. 12 provides a general overview of the WRF-Chem output for the LA basin and surrounding areas at the closes time to the ozonesonde profile (28 May 1:00 UTC). The 2500 m horizontal cross-

section presented in Fig. 12a reveals a fairly large plume of LA pollution, characterized by high ozone and high anthropogenic CO, being transported eastwards. The A-A' cross-section shows an example of the typical mountain venting (or mountain chimney) effect, which has been previously identified as the main mechanism by which LA basin pollution is transported over the San Gabriel mountains and into the free troposphere (Langford et al., 2010). About 2.5 km a.s.l., a slight decrease in the ozone mixing ratio is visible over TMF on both cross-sections, which corresponds to the same undercutting dry and ozone-

poor layer previously discussed. The surrounding Phelan and Crestline stations exhibited an opposite behavior to TMF, with the surface ozone greatly exceeding the EPA threshold and measurements showing generally higher ozone than forecast (Fig. A2).

## 4.2 3 July 2020: A deep stratospheric intrusion event

This case study, based on measurements conducted by TMTOL on 3 July 2020 between 4:50 and 17:50 UTC, illustrates the
effect of a deep stratospheric intrusion on the surface ozone concentration at TMF. Since deep stratospheric intrusions typically
occur during spring and not necessarily significantly affect the surface ozone concentration, the case study presented in this
section represents a rarity in both ways. Additionally, this case study illustrates the challenges of forecasting the impact of SIs
on surface ozone concentration. In particular, the difficulties associated with an accurate representation of the entrainment into
the nocturnal surface layer in complex terrain.

Figure 13a presents an overview of the forecast WRF-Chem ozone, anthropogenic CO (grey-shaded contours), PBL height
(dashed black), and WACCM stratospheric ozone contribution (hatched contours) for the period comprehended between 2 July
2020 12:00 UTC and 4 July 2020 0:00 UTC. The forecast, which motivated an extended TMTOL run, starts with moderate
ozone enhancement in the PBL during 2 July with an associated increase of anthropogenic CO, suggesting transport from the
LA basin area. After the collapse of the PBL, a persistent layer of high ozone (about 70 ppbv) is visible between the ground and
3.5 km a.s.l., with a small drop around 3 July 6:00 UTC. Until that point, the forecast can be seen as a typical case of LA basin
pollution transport, with ozone being injected into the free troposphere after the collapse of the PBL. Nevertheless, after 6:00
UTC, a second increase of the ozone in the lower troposphere is visible, which later extends into an ozone-rich stratospheric
air tongue with ozone levels exceeding 70 ppbv. The stratospheric origin of this enhancement is supported by the co-located
low relative humidity from WRF-Chem (Fig. 14b), as well as the stratospheric ozone tracer from WACCM, which shows a
contribution of over 30 ppbv on 3 July between 4:00 and 13:00 UTC at about 2.5 km a.s.l.. After 13:00 UTC, and as the PBL
starts to grow, the influence of the SI starts to decrease and a new transport wave of LA basin pollution takes over as the main
ozone driver in the PBL.

While the models presented in Fig. 13a are in qualitative agreement with the TMTOL measurements (Fig. 13b), measure-
ments show a shallower layer with a much stronger enhancement in the ozone levels below 3 km a.s.l. and a less defined
stratospheric air tongue than the simulation. During the whole measurement period, the observed layer was characterized by
ozone levels of over 100 ppbv (as opposed to the 70 ppbv forecast by WRF-Chem) and low aerosol load levels (Fig. 13c).
Another significant difference is related to the surface influence of the SI. Figure 13d presents a time series of forecast and
measured surface and 100 m a.g.l. ozone mixing ratio, as well as the measured relative humidity at TMF. While the WRF-Chem
simulation forecast an almost constant 50 ppbv surface ozone level, compatible with nighttime background conditions, the 49i
Ozone Analyzer shows increasing ozone values after sunset, reaching 90 ppbv between 9:00 and 11:00 UTC. While looking
at the 100 m a.g.l., no significant difference was observed between the first valid TMTOL data point (100 m a.g.l.) and the
surface, while the WRF-Chem model showed a gradient of about 15 ppbv between surface and 100 m a.g.l.. This difference is
better depicted in Fig. 14, where ozone, relative humidity and potential temperature as forecasted by WRF-Chem are presented
for three different times. The first profile corresponds to the pollution transport event during the late afternoon (A, 0 UTC),
with a 500 m deep PBL characterized by an ozone concentration of over 70 ppbv, a relative humidity of almost 30 % and a
moderately strong temperature inversion at its top. Just above the PBL, and characterized by a relative humidity of 10 %, we

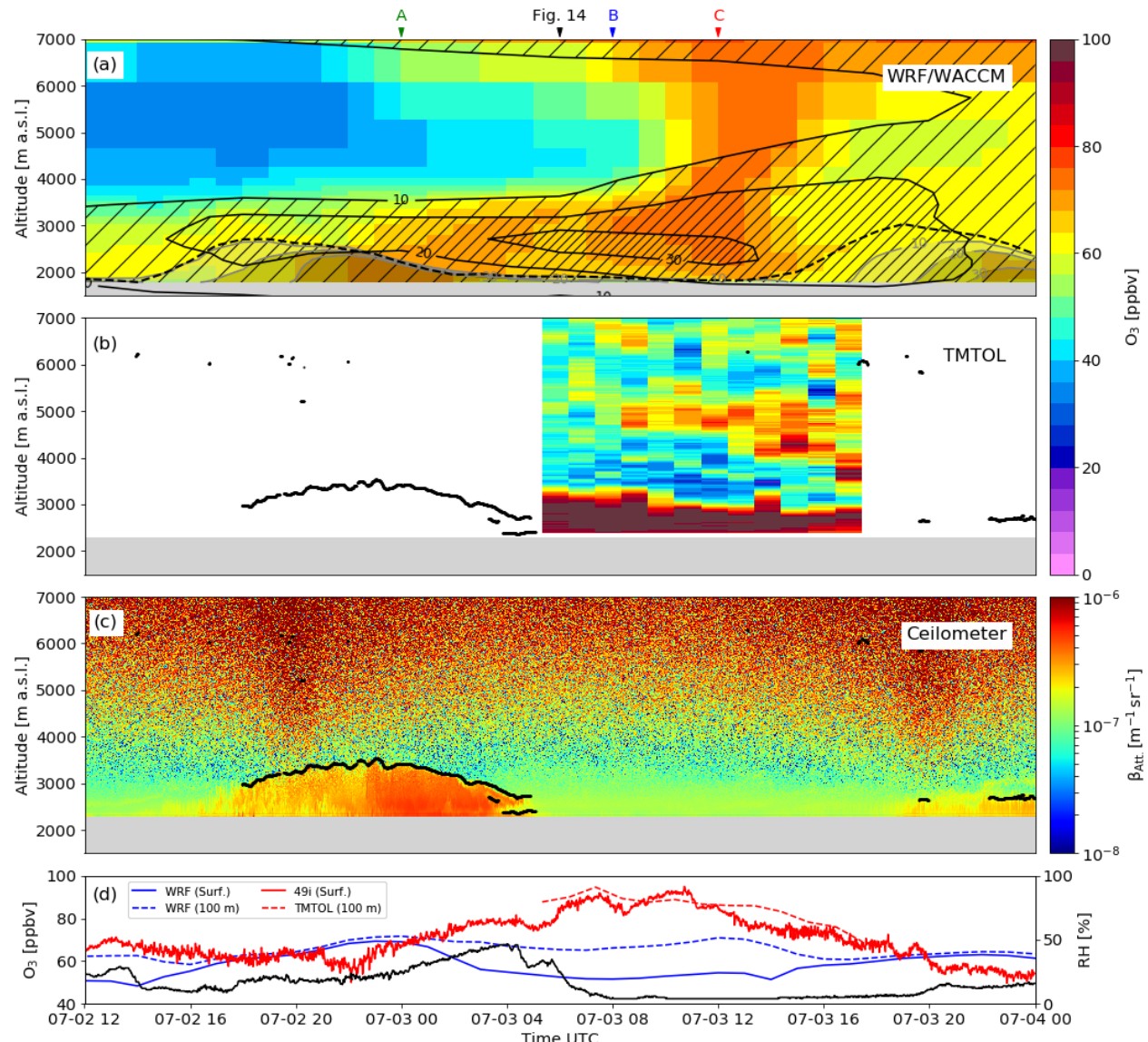

**Figure 13.** Overview of the model outputs and measurements over TMF between 2 July 12:00 and 4 July UTC. Panel descriptions are the same as the ones shown for Fig. 10. The time of the profiles shown in Fig. 14 are shown as red arrows.

can see the SI influence forecasted by WACCM and shown in Fig. 13a. As the PBL collapses and the SI approaches the surface (B and C), a strong temperature inversion develops near the ground, which inhibits mixing of ozone from the SI and limits its impact in the surface.

As in the previous case study, the same variables shown in Fig. 13a for 3 July 6:00 UTC are presented in Fig. 15 for a constant altitude of 2.5 km a.s.l. around the LA basin area as well as for the two cross-sections defined in Fig. 7.

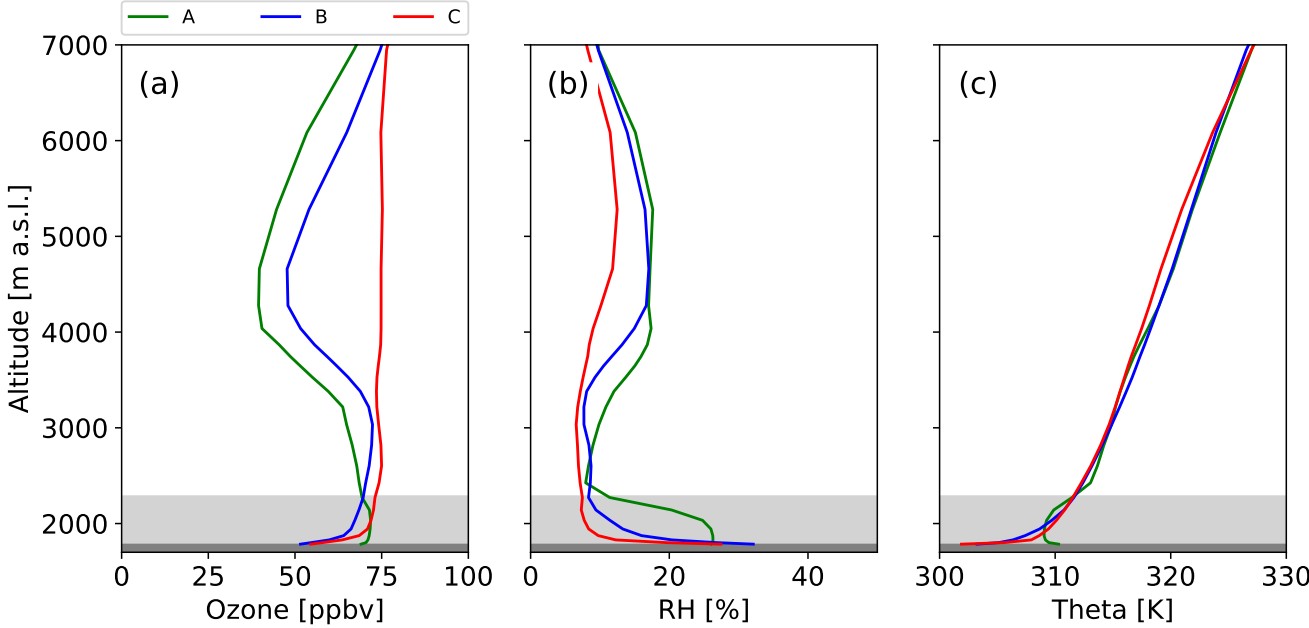

**Figure 14.** WRF-Chem ozone, relative humidity and potential temperature over TMF at times A (green, 0 UTC), B (blue, 8 UTC) and C (red, 12 UTC) indicated by red arrows in Fig. 13a top. The actual TMF elevation (light grey shaded) is shown together with the model elevation (grey shaded).

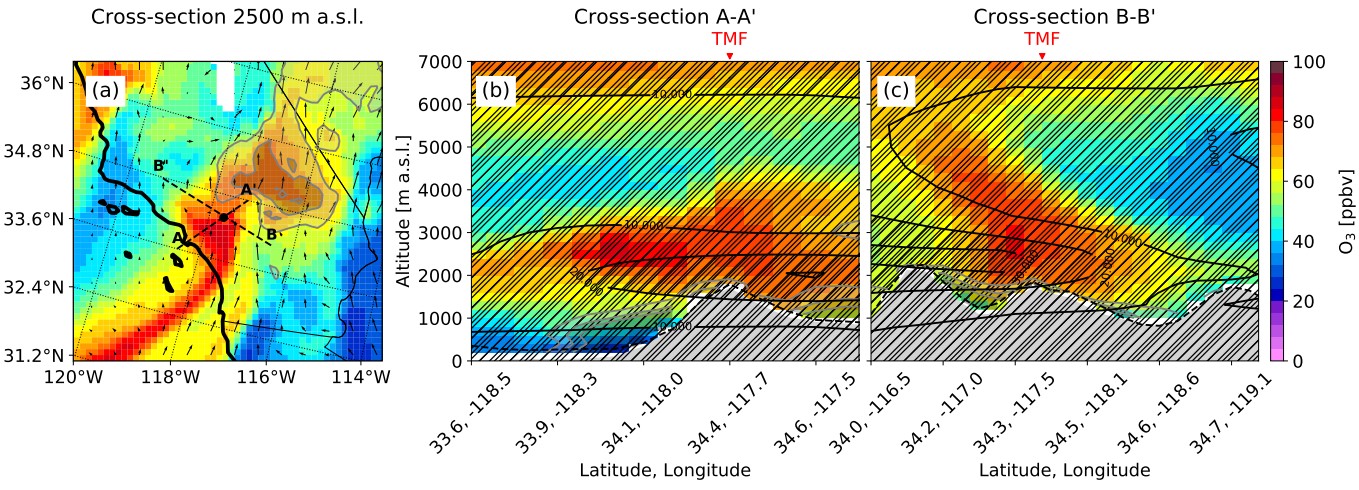

**Figure 15.** Horizontal and vertical cross-sections of WRF-Chem and WACCM forecast outputs for 3 July 6:00 UTC. Panel descriptions are the same as the ones shown for Fig. 12.

The constant altitude cross-section (Fig. 15a) shows a well defined high ozone filament south-west of TMF associated with high speed south winds and low humidity (not shown). Slightly north from TMF, a second plume of high ozone and anthropogenic CO (LA basin pollution) can be seen being displaced by this high ozone filament from stratospheric origin. The cross-sections presented in Figs. 15b and 15c provide an additional overview of the stratospheric tongue geometry as well as the stratospheric contribution forecast by WACCM (hatched contours). The A-A' cross-section shows also very low ozone over the LA basin area, as usually found during nighttime and suggests that little or no impact of this SI was forecast.

### 4.3  11-15 June 2020: A combined case

In this case study, an extended TMTOL run conducted between 10 June 15:20 and 15 June 7:10 UTC is analyzed and compared to the WACCM and WRF-Chem forecasts. In contrast to the previous two case studies, this measurement period provides a more comprehensive and complex picture of the multi-day evolution of the different processes affecting the surface ozone levels at TMF.

During the first two days of this case study, the WRF-Chem and WACCM simulations (Fig. 16a) forecast little stratospheric influence as well as considerable transport of LA basin pollution during 11 June. TMTOL ozone retrievals (Fig. 16b) and the ceilometer-derived PBL height (Fig. 16b, black) show very good qualitative agreement with the WRF-Chem simulations during 11 June, with larger differences observed during 12 June. During 11 June, a deep PBL reaching 4.5 km a.s.l. (11 June 0:00 UTC) can be inferred from both the forecast and the observations (TMTOL and Ceilometer), with enhanced ozone, anthropogenic CO and aerosol load. In the case of the WRF-Chem simulation, the ozone mixing ratio associated with this transport event was in the order of 70 ppbv at 100 m a.g.l. (Fig. 16d), which agreed well with TMTOL observations. Nevertheless, the simulations also show a very strong gradient in the lower 100 m, which resulted in an underestimation of the impact of this LA basin plume at the surface and prevented the forecasting of the exceedance as it finally happened. In the case of the 12 June forecast, the differences with the TMTOL, ceilometer, and surface instruments were larger compared to the previous day. WRF-Chem simulations forecast only mild ozone transport, while TMTOL and the ceilometer measurements show ozone mixing ratios over 80 ppbv associated with a strong aerosol load. As in the previous day, the surface ozone records show a second exceedance, while the forecast underestimated the surface impact by about 20 ppbv. This underestimation of the forecasted ozone is likely related to a difference between the forecasted and the actual wind fields. The forecasts show a fairly constant wind direction of 210 degrees (transport from central LA) after 11 June 22:00 UTC, while the measured wind direction at TMF was about 180 degrees (transport from the Fontana/San Bernardino area, south of Crestline). This difference corresponds to two different transport regimes as can be seen in Fig. 4. Another remarkable feature can be seen on 12 June 13:00 UTC (indicated as 'See text' in Fig. 16c), when the irruption of a thin near-ground layer with high aerosol content made the surface ozone measurements climb briefly up to little under 100 ppbv, while keeping the 100 m a.g.l. TMTOL measurements unaffected.

Starting on 12 June 12:00 UTC, an increase in the stratospheric ozone contribution can be observed developing in WACCM and WRF-Chem forecasts above the PBL. This ozone enhancement, also visible in TMTOL profiles, is followed by two descending ozone-rich stratospheric air tongues. TMTOL observations indicate a good qualitative agreement of the model regarding these two deep intrusions, with some differences on their timing, spatial evolution, and ozone mixing ratio. Shortly

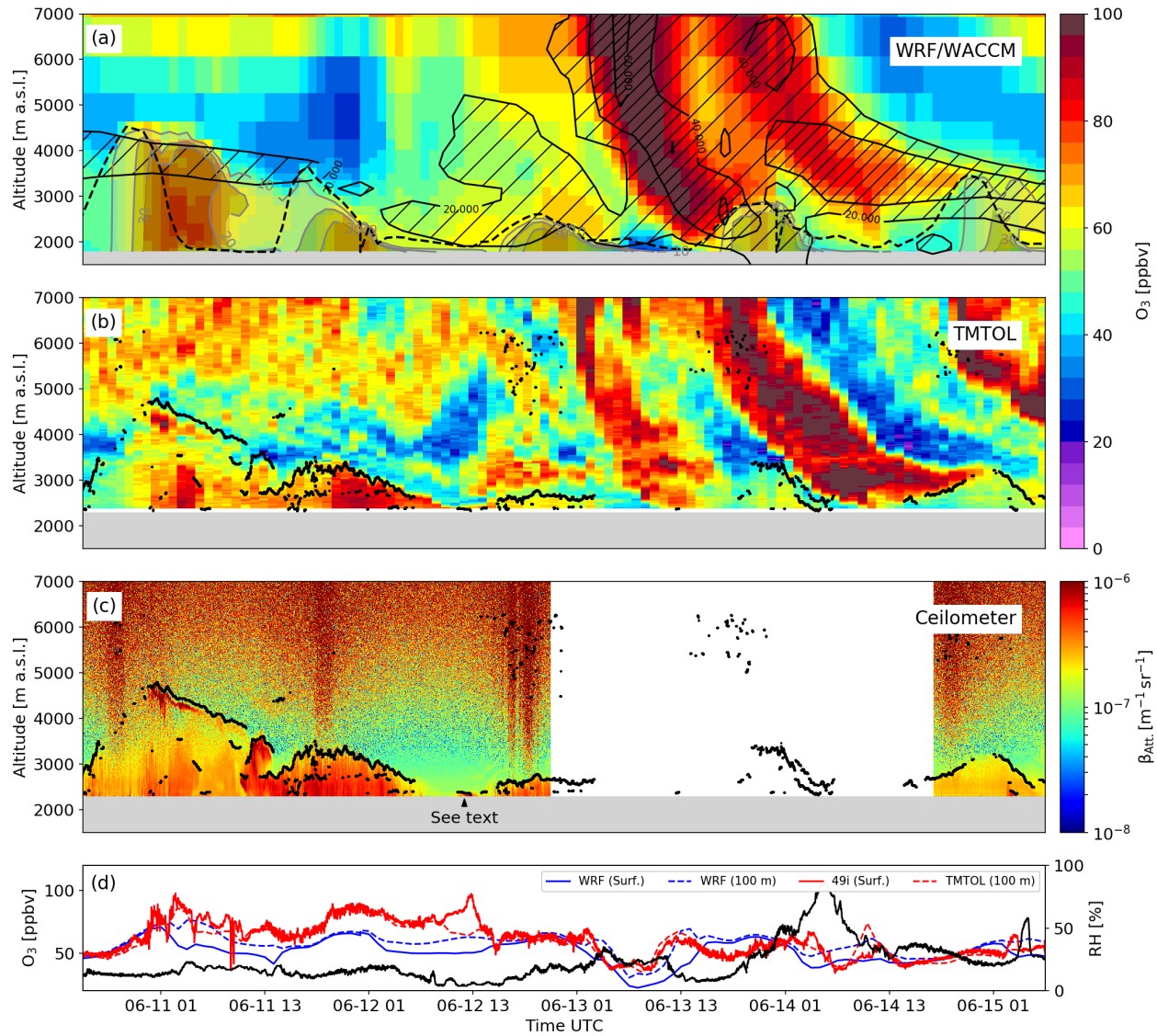

**Figure 16.** Overview of the model outputs and measurements over TMF between 10 June 15:20 and 15 June 7:10 UTC. Panel descriptions are the same as the ones shown for Fig. 10.

after the first intrusion is seen appearing at 7 km a.s.l. (13 June 3:00 UTC), and coincident with the collapse of the PBL, a relatively sharp decrease in the surface ozone mixing ratio from 70 ppbv to 50 ppv can be seen in Fig. 16d. This decrease in the surface ozone concentration, associated with an increase in the relative humidity, resembles the case study presented in Chouza et al. (2019) and suggest low-level transport of marine air as the source of it. By 13 June 13:00 UTC, the stratospheric intrusion reaches the surface over TMF, causing a decrease in the surface relative humidity and an increase in the surface ozone mixing

ratio. The WRF-Chem ozone mixing ratio at 100 m a.g.l. shown in Fig. 16d shows a very good agreement with the TMTOL retrieval at that altitude, while the forecast of the surface ozone shows a considerable underestimation likely associated with the growth of the PBL and limited downward entrainment. In a similar way, the second intrusion is preceded by a decrease in the surface ozone and a strong increase of the relative humidity, associated with marine air transport (back-trajectories not shown). As the SI approaches the surface, the relative humidity decreases, and the surface ozone increases. While the near-ground TMTOL measurements (Fig. 16d) show an ozone mixing ratio peak of about 75 ppbv at the time the intrusion is at its minimum altitude over TMF, the surface ozone and the WRF forecast show only very weak signs of it. Finally, as the PBL starts to grow, the intrusion is pushed up generating a strong gradient between the PBL and the SI with no evident signs of ozone enhancement in the PBL due to entrainment neither in the model or the surface measurements.

The ozone monitoring station in central LA (Fig. A3) did not record any obvious increase in surface ozone associated with any of the two SI events. For the second SI event, the ozone concentration actually dropped more abruptly than forecast, likely related to the onset of the cold front and associated reduced photochemical activity. The Crestline surface monitoring station (the closest high-elevation site to TMF) did show an episodic increase in surface ozone concentrations a few hours before the increase observed at TMF. Surface ozone forecast at the surrounding stations followed the general trend of the observations, but failed to reproduce the SI related increases at the TMF and Crestline station; likely as a result of insufficient entrainment into the nocturnal surface layer. These episodes resemble a case study presented in Langford et al. (2012), where another deep SI was determined to be responsible for an ozone threshold exceedance at the Joshua Tree National Park during 28 May 2010, while the rest of the stations in the LA basin showed a decrease in the ozone concentration as a result of decreased photochemical activity associated with the passage of a cold front.

## 5   Summary, conclusions and outlook

In the first part of this work, surface measurements conducted at TMF were used, in combination with WRF-Chem and WACCM simulations, to provide an overview of the near-surface ozone and PM10 characteristics at TMF. The results revealed a large number of days with ozone levels exceeding the National Ambient Air Quality Standards at TMF during late spring, summer and early fall, with the maximum during June. During this period, the surface influence of stratospheric intrusions is modeled to be at its minimum. This result, in combination with the large concentrations of anthropogenic CO forecast by WRF-Chem at TMF, suggests that LA basin pollution plays a dominant role in these exceedance events, regardless of season. Backward-trajectories indicate that the surface ozone at TMF is highly sensitive to the wind regime, with the highest ozone typically observed associated with eastward transport from the Santa Clarita/San Fernando Valley urban area.

Making use of the recently enhanced TMTOL measurement capabilities, 726 vertical profiles of ozone measured during TROPOMI overpasses (noon) and shortly after sunset were used to conduct an evaluation of the ACOM WRF-Chem air-quality forecast and WACCM over the period comprehended between May 2019 and September 2020. The comparison revealed a fairly good agreement in the PBL, with WRF-Chem generally overestimating the ozone concentration by less than 25%. Above the PBL, a high ozone bias was observed, increasing with altitude and reaching about 75% at 15 km a.s.l.. This bias appears to

be season and time-independent, and might be partially responsible for the bias observed in the PBL during noon and after sunset. Further measurements during late afternoon, when ozone concentration is expected to be mainly driven by pollution transport from the LA basin, would be required in order to further investigate the impact of the upper troposphere bias in the PBL ozone levels. The comparison with WACCM, used to initialize each of the WRF-Chem runs, revealed a very similar altitude-dependent bias over TMF, Trinidad Head and Boulder, suggesting that the bias is carried over to WRF-Chem from the WACCM runs.

Additionally, three case studies showing stratospheric and LA basin driven exceedances were discussed in light of the WRF-Chem and WACCM stratospheric ozone forecast. A good agreement was observed between TMTOL measurements and simulations concerning the different mechanisms driving near-surface ozone, including LA basin transport through mountain slope effect and stratospheric intrusions. For these case studies, the most distinctive differences between the observations and model regarding the surface ozone levels were observed to be related to near surface temperature inversion that inhibit the down-mixing of the ozone. Additional measurements would be required to evaluate the frequency of these discrepancies.

While the results shown in this work point to the mountain chimney effect and associated LA basin transport as the main mechanism controlling the abundance of ozone and other pollutants in the PBL, residual layers, and lowermost troposphere over TMF, additional datasets including surface measurements of CO, $NO_2$ and stratospheric tracers (like beryllium-7) would provide additional support to this conclusion and allow a quantification of the actual stratospheric contribution to enhanced ozone levels. Additional lidar measurement capabilities, allowing simultaneous lidar measurements in the LA basin area, TMF, and Mojave desert would provide further evidence of regional transport as well as a better understanding of the impact of these elevated plumes on downwind ozone monitoring stations like those deployed in Mojave and Joshua Tree national parks. Furthermore, the deployment of such lidar systems in a coastal environment upwind from the L.A. basin, and where the influence of local sources is limited (e.g. on the Channel Islands), would allow to better characterize and quantify the tropospheric ozone background conditions of air entering the west coast of the continental United States (Oltmans et al., 2008).

*Data availability.* The TMTOL data sets used in this study is publicly accessible at https://www-air.larc.nasa.gov/missions/TOLNet/data. html. ACOM WRF-Chem forecast output is accessible at https://www.acom.ucar.edu/firex-aq/FIREX-AQ/Evaluation/TOLNet/Wrightwood_ CA/. ACOM WACCM dataset can be requested at https://www.acom.ucar.edu/waccm/download.shtml. Ozonesondes from Trinidad Head, CA and Boulder, CO can be downloaded from ftp://aftp.cmdl.noaa.gov/data/ozwv/Ozonesonde/ For auxiliary datasets, please contact the authors.

*Author contributions.* Fernando Chouza prepared most of the manuscript and the statistical comparison with the rest of the datasets. Thierry Leblanc is the principal investigator of TMTOL and provided support in analysis of the TMTOL data. Gabriele Pfister, Rajesh Kumar and Carl Drews provided the WRF-Chem forecast used in this study. Simone Tilmes and Louisa Emmons provided the WACCM simulations used in this study. Mark Brewer and Patrick Wang provided technical support for the collection of the data included in this work. Sabino

Piazzolla provided the ceilometer and PM10 data. Matthew Johnson provided information used to decide when to run TMTOL (case studies) and input on the lidar-model intercomparison. All co-authors provided feedback on the manuscript.

*Competing interests.* The authors declare that they have no conflict of interest.

*Acknowledgements.* The research was carried out at the Jet Propulsion Laboratory, California Institute of Technology under a contract
with the National Aeronautics and Space Administration (80NM0018D004). The development of the WRF-Chem forecasting system was supported by the NASA grant 80NSSC18K0681. We would like to acknowledge high-performance computing support from Cheyenne (doi:10.5065/D6RX99HX) provided by NCAR's Computational and Information Systems Laboratory, sponsored by the National Science Foundation. We acknowledge the use of the WRF-Chem preprocessor tool anthro_emis provided by the Atmospheric Chemistry Observations and Modeling Laboratory (ACOM) of NCAR. The National Center for Atmospheric Research is sponsored by the National Science
Foundation. We acknowledge NOAA ESRL/GML for the ozonesonde data used in this study.

## Appendix A: Appendix A

The following figures provide supporting information for the discussions presented in Sec. 3.2, Sec. 4.1, and Sec. 4.3.

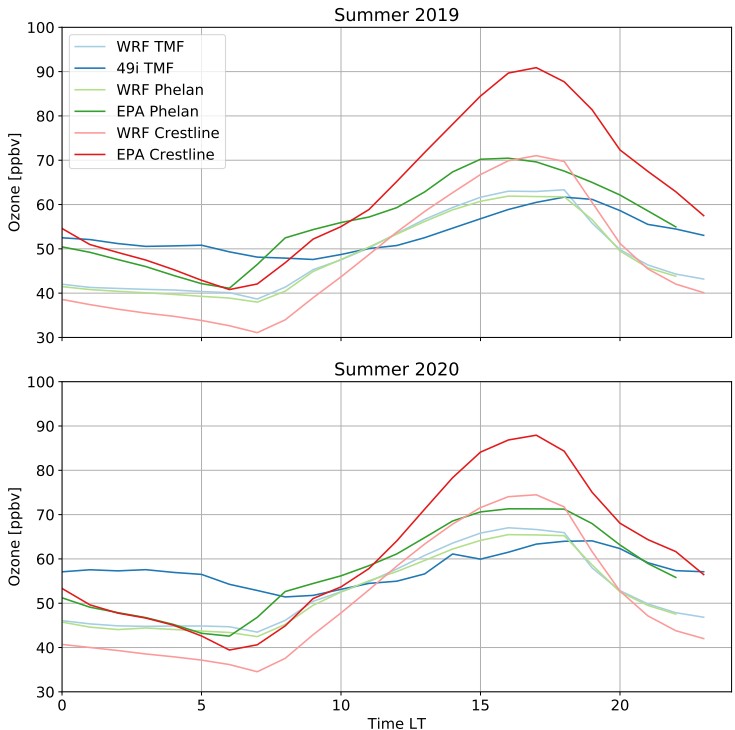

**Figure A1.** Mean ozone daily cycle at TMF and nearby stations (Phelan and Crestline) together with the corresponding WRF-Chem output for summer 2019 and summer 2020.

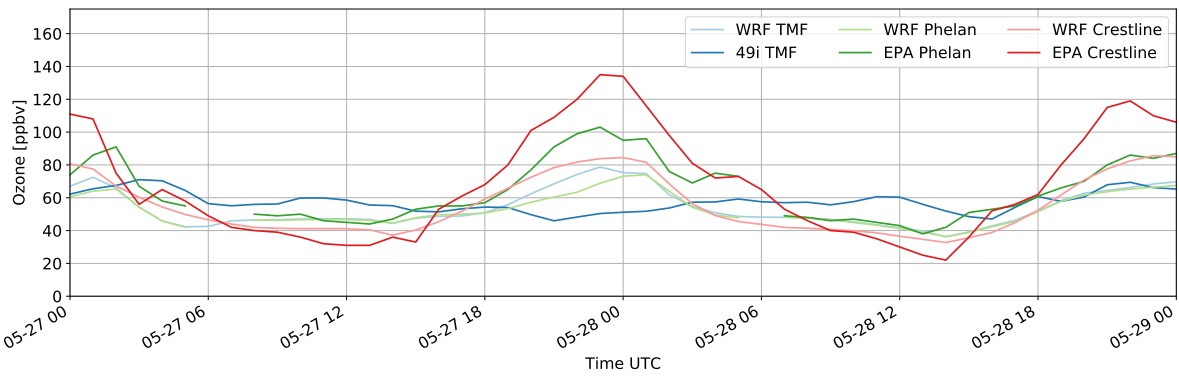

**Figure A2.** Forecast and measured surface ozone concentration at TMF and nearby stations (Phelan and Crestline) for the period comprehended between 27-28 May 2020.

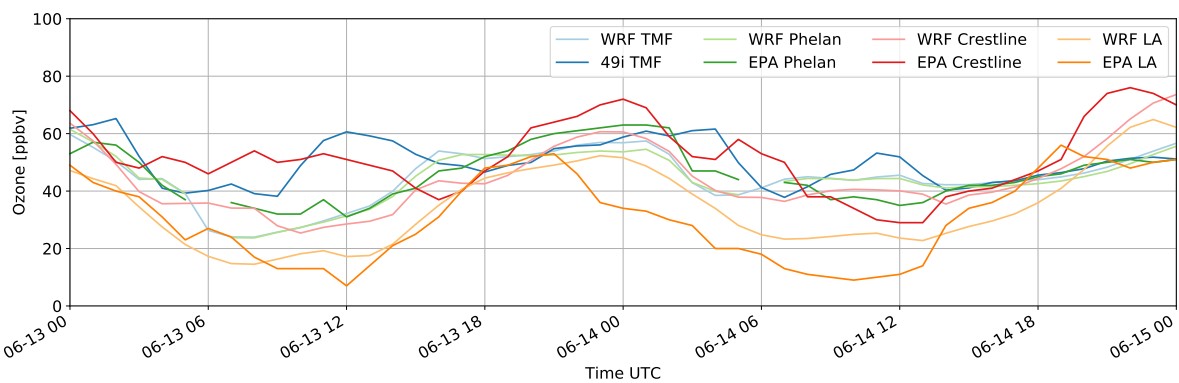

**Figure A3.** Forecast and measured surface ozone concentration at TMF and nearby stations (Phelan, Crestline and LA) for the period comprehended between 13-15 June 2020.

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
