# Peer review of "The impact of Los Angeles basin pollution and stratospheric intrusions on the surrounding San Gabriel Mountains as seen by surface measurements, lidar, and numerical models"

_Atmospheric Chemistry and Physics, 2020_

## Referee Comment (RC1) · Anonymous Referee #1 · 12 Jan 2021

Chouza et al. present an interesting study comparing ozone lidar measurements from the Table Mountain Facility with model simulations from the NCAR WRF-Chem and WACCM models. The study includes an overview of the measurements from May 2019 through September 2020, and presents three case studies from this period describing a stratospheric intrusion event, a regional pollution transport event, and a third event with both influences. The analysis is thorough and compelling, but I think the authors should also use measurements from the extensive network of ozone monitors in the

[Figure]

Los Angeles area to provide more context for their results. This would better link these mountaintop measurements to the issue of regional ozone attainment they invoke in both the introduction and conclusions.

I have a number of minor comments and suggestions for improved readability.

P2, L46. "...near-surface measurements carried out at..."

P3, L63. "...northwest of Wrightwood..", "The site hosts..."

P3, L64-67. It is never clearly stated that the Chen study was based on DOAS. Perhaps start with "Differential Optical Absorption Spectroscopy (DOAS) measurements by Chen et al..."

P3, L67. "Similar conclusions have been reached..."

P4, Table 1. Is the temporal resolution of the TMTOL really 1 hour or is that just the integration time used for the comparisons?

P4, L86 and Figure 2. Is there an explanation for why the TMTOL values are consistently higher than the UAV values below 75 ppb? Did the UAV carry a standard ECC ozonesonde? Also, what is the relevance of showing the "Days since first validation"? Is this meant to somehow account for the one outlier?

P7, L164. Would the authors care to comment on the clear seasonal shift between 2012-2014 (highest O3 in May and June) and 2017-2020 (highest O3 in July and August) in Figure 3a?

P8, L181 and Figure 4. Are the seasons defined here as Fall (SSO), W(DJF), Sp(MAM), and Summer (JJA) or by equinox/solstice?

P8, L183 and below. "forecast" is the preferred form of the verb-not "forecasted".

P8, L186. Are the shifts in wind direction consistent with the lower resolution of the model topography?

P8, L195. What was the resolution of the meteorology used for the HYSPLIT back trajectories?

P8, L206. "The trajectories corresponding to the 210-240° prevailing winds..."

P8, L207. "...with respect to the 250-300° back trajectories..."

P9, Figures 3b-3d. These plots are confusing. What statistics are represented by the box-and-whisker plots? I assume the shading refers to 2019 and 2020, but this is not explicitly stated. What do the dark gray diamonds represent? By the way, the plotted symbols are time series and not "scatter plots".

P11, L211. The discussion here skips back from Figures 5 and 6 to Figure 4 without warning. It took me a while to figure out the authors were referring to Figure 4 in the paragraph that followed. Perhaps revise to "During summer 2019, PM10 observations and forecast (third and sixth rows of Figure 4)...".

P11, L213. "A difference is observed for ENE..."

P11, L229. "...focused on the free troposphere..."

P11, L214. Were there any significant wildfire influences in the Fall of 2019?

P11, L217. The Met One 212 has a lower size cut at 0.3 $\mu$m and thus may be excluding the smaller particles in the model analysis.

P12, 13. Figures 5 and 6. The gray back trajectories are hard to see in some of the plots. Perhaps use heavier white or magenta lines?

P16, L279+. In my opinion, it would be better to switch sections 4.1 and 4.2 and describe the pollution transport event first since this is the more typical event. That would help to put the TMTOL and ceilometer measurements from Figure 9 in better context.

P16, L284. This sentence is awkward and could perhaps be phrased better.

P17, L295. The WRF-Chem RH isn't shown.

P18, F9 caption. (a) The CO scale is missing. (b) Please note that the PBL height is from the ceilometer measurements in (c). The ozonesonde profile mentioned in the caption does not appear to be in the plot. (d) The plot is already complicated, but it would be useful to see the WRF-Chem RH since it is mentioned in the text.

P19, L340. Again, it would be useful if the TMF surface measurements were compared with the regulatory measurements, particularly those from particularly Phelan and Crestline.

P21, L357. "...the closest time to the ozonesonde..."

P23, L281. The high aerosol content of the irruption is not obvious from the ceilometer measurements.

P24, L396. This case can be contrasted with that described in Langford et al 2012* where the descent of a deep SI also caused surface O3 to decrease in the Los Angeles Basin. In that case, however, the surface RH also decreased as drier air from aloft displaced local pollution.

P24, L406. "This result,..."

* Langford, A. O., J. Brioude, O. R. Cooper, C. J. Senff, R. J. Alvarez, R. M. Hardesty, B. J. Johnson, and S. J. Oltmans (2012), Stratospheric influence on surface ozone in the Los Angeles area during late spring and early summer of 2010, J. Geophys. Res., 117, D00V06, doi:10.1029/2011JD016766.

---

## Referee Comment (RC2) · Anonymous Referee #2 · 15 Jan 2021

The manuscript submitted by F. Chouza and co-authors concerns the analysis of ozone measurements at the Table Mountain (TMO) California station. The objectives are multiple (1) to demonstrate the contribution of new measurements at low altitudes of the TMO lidar for the analysis of regional pollution (2) to analyze the respective role of stratospheric intrusions and regional pollution on ozone threshold exceedances (3) an evaluation of the performance of WRF-CHEM model ozone forecast for the analysis of pollution episodes.

[Figure]

The paper contains many interesting results based on a joint analysis of observation and modeling. It certainly deserves a publication in ACP. My main criticism will be on the readability of the objectives and the overall coherence of the paper. The authors are probably trying to meet too many objectives in the same paper. The evaluation of the WRF CHEM forecast seems to be both the most original objective considering the previous publications on the analysis of ozone measurements in the Los Angeles area and a good approach to ensure consistency in the paper. This could be improved by (1) a better presentation of the objectives in the introduction (2) add a discussion of model/measurement differences in section 3.1, otherwise the results presented will be difficult to use (3) Focus section 4 on the evaluation of model performance for the three case studies. The latter could be seen as examples to discuss the differences identified in section 3.2 using the 726 lidar profiles. Indeed the identification of the 3 high surface O3 drivers at TMF based on section 4 is not really new (see previous papers by Cooper, Langford, Lin) and the same results could be very useful if the discussions focus on the model forecast performances to represent theses processes.

Detailed remarks

Line 37 and 51: Relevant objectives are provided here and the introduction generally misses a list of these objectives in addition to the list of tools used.

Line 40-44 : While the representativity of a mountain top station to characterize PBL or free tropospheric air is a very important point, it is not clear how the paper address this question in section 3.1 or in section 4. If some results address this question they should be mentioned in section 3 or 4.

Line 175 : Scatter of WACCM stratospheric tracer on O3 exceedance days are a little hard to read in figure 3.b but it seems that the mean is always higher than the median or even than the upper interquartile of the stratospheric tracer for the overall data set. So is it true to say that the stratospheric intrusion plays a limited role based on this unique feature ? I am not sure I understand the differences between the black and

color dots in Fig. 3. On the other hand it is true to say that the O3 exceedance days with RH>50 % are often observed. So interpretation of the WACCM forecast is not straightforward. May be the age of the stratospheric tracer could help.

Line 179 : Make a new section here because there is no obvious link between the analysis of the O3 exceedance days shown in Fig. 3 and the new question discussed in Fig. 4 where the wind and diurnal cycle dependency of surface and model are shown.

Line 190 : I agree it is a nice result of this study. However one could expect some hints about of the model difficulties to reproduce this diurnal cycle. Is a well known feature or specific to the TMO data set ?

Line 191 -201: The authors give a great importance to the distribution of ozone in the prevailing wind sector but why do you expect an ozone maxima for this sector ? Indeed according to the ozone distribution shown in figure 5, the ozone max are found west and south of TMO. So the TMO surface ozone max is obviously found for the western and southern wind sectors. I propose to simplify the discussion line 191-201 and use Figure 5 to explain why the observed ozone maxima are for the western and southern sectors. Then the second level of discussion is the comparison of the ozone/wind dependency for the observations and the forecast, which is quite good.

Figure 5 : add TMO position on this figure, trajectories are also hard to read and it is not clear why they are different for top, middle and bottom panel

Section 3.1 A summary about the main findings about this surface measurement/model comparison must be given at the end of the section: accuracy of model ozone field around TMO, biais in diurnal variation, accuracy of ozone titration in the model.

Section 3.2 It is a very interesting section and there is little published work where many lidar profiles are statistically compared with regional-scale CTM simulations. However the discussion is not very developed and the lidar vertical resolution is probably not

the main driving factor. While large differences due to small time or spatial shift of the simulated ozone layers above TMO are expected near the tropopause or at the PBL top, it is unexpected to find 30-50 % differences at 7.5 km. A small discussion about possible reason for this overestimate of the forecast could be added. Was this feature already observed during model evaluation in the free troposphere ? Is it specific to TMO and why ? The differences seems also worse during nighttime (except in summer 2020) while the lidar accuracy is less for daytime observations. Do you have an explanation for this ?

Section 4.1. It is indeed an interesting case study to demonstrate the role of stratospheric intrusion for increasing surface ozone measurements at a mountain top station. However it is not a new result which has been reported in several publications. So a discussion is missing either to compare this new case study with previous ones (Bonasoni 2000, Trickl 2019, Knowland 2017) or to explain why the model did not forecast a surface ozone enhancement (line 285 and 300).

Line 308 Can the authors use their detailed spatio-temporal analysis of the measured and forecasted ozone field in Figures 9 and 10 in order to discuss the differences in the time series at TMO ? Can a model shift along BB' explain the observations ? Can such a shift explain why the model is too low at the surface or is the model mixing too low in the PBL ?

Figure 10c The dashed line above mountain top is not defined. Is it the PBL top like in figure 9 ? Add a vertical line in Figure 9 at the time corresponding to the horizontal/vertical cross-sections shown in Fig.10. Similar comment for fig. 13 and 11.

Section 4.2. This case is indeed interesting for the model assessment because the large scale upper tropospheric ozone feature are weaker than case 4.1 and small shift of the modeled ozone field will not easily explain the lidar/model difference.

Line 339. Does this kind of model overestimate correspond to the statistical positive bias between model and TMTOL near 3 km which is discussed in section 3.2 ?

Line 341 It is worth mentioning that the forecast daily maximum is earlier in the model as previously shown in section 3.1 using the surface observations. It seems to be a permanent feature of the forecast.

Line 378 What is the reason for the model performing less for this case study ?

Line 344 Is mixing too high in the model close to the ground ?

Line 413 Although the model/TMTOL differences are lower than 20% within the PBL, it must be stressed that there is a model positive bias which might be related to the large differences observed in the free troposphere between 4 and 7 km.

---

## Referee Comment (RC3) · Anonymous Referee #3 · 27 Jan 2021

The submitted manuscript submitted by F. Chouza and co-authors concerns the analysis of ozone measurements at the TOLNet/NDACC JPL/Table Mountain (TMO) California station.

The manuscript focuses on the impact of Los Angeles basin pollution transport and stratospheric intrusions on the surface ozone levels observed in the San Gabriel Mountains is investigated based on a combination of surface and lidar measurements as

well as WRF-Chem (Weather Research and Forecasting with Chemistry) and WACCM (Whole Atmosphere Community Climate Model) model runs.

This manuscript poses a variety of central thematically relevant questions that could be explored further. The introduction hints at but does not investigate increasing the accuracy of detangling the high-altitude sites (generally use for long term background/clean trends) and their respective local ozone episodes. There is comparisons with model data, but there really isn't a way to extract conclusions that are satisfying in my opinion. The manuscript also begins to describe the ozone NAAQS and ozone exceedance episodes from a regulatory perspective, but does not really go beyond a simple description and table which is not referenced in the case studies later in the manuscript. Many of this has been done previously by the papers referenced within the document (e.g. Cooper, Lin, Langford).

I suggest the authors reorder the manuscript to re-emphasize the new case studies and comment on their regulatory implications in a discussion section. What is clear is the rich and novel data set in the latter half of the manuscript, especially the 2020 episode. This is very appropriate for this journal with a description of new science learned from this case study. This shows clear evidence of the upslope flow/LA basin impact followed directly by a stratospheric intrusion.

L23 – Carbon monoxide is not central to the photolysis to generate ozone. CO is certainly a useful tracer. Did you mean OH radical? L25 – for this site location, is transport notionally from the LA Basin in a regional sense or inter-continental transport on a synoptic scale?

L45 – a "few", is there a way to more quantitatively define this? L75 – is this lidar also part of the NASA TOLNet network? I assume so considering the work in LeBlanc et. Al. 2018.

Figure 2: What is trying to be conveyed in this z axis? That there is negligible instrument drift? Would a time series be more appropriate? Are the 90ppbv measurements

taken during the lidar data shown in subsequent sections? Having a 100m to surface comparison would also be useful to illustrate the gradient on select days.

L118 – If the model does not include stratospheric chemistry, is this really an appropriately rigorous evaluation?

Figure 5 – The back-trajectories are difficult to see. Consider alternative approach. Since most of the trajectories are in the same general cardinal direction, why not use a vertical image instead to illustrate the particle dispersion and upslope/mountain chimney effects?

Figure 8 – It is challenging to see a clear result from this figure. Is this an emphasis on showing seasonal changes in ozone or an evaluation of the model? Is the model just not brining enough stratospheric ozone into the troposphere? There is a systematic bias above 7500km. Could this figure be replicated but for a time series of Tropospheric ozone Columns? This product may also be relevant to the upcoming geostationary satellites.

Figure 12 – This is a careful examination of the model and geophysical features. Well done.

Figure 14 – This is the clearest evidence of a LA Basin pollution transport followed by a stratospheric intrusion reaching the western US I have seen. Very appropriate for the journal and the community needs to be aware of these measurements. This should be re-emphasized as the central part of this manuscript.

---

## Author Comment (AC1) · 26 Feb 2021

**General comments**

We would like to thank the three reviewers for their insightful comments that helped to greatly improve this work. The manuscript was modified to address the requests from the reviewers, while keeping the original objectives of the work.

Matthew Johnson was added to the co-authors list to give him credit for providing information that allowed to conduct measurements during the case studies shown in this work. He also provided insightful comments on the WRF-Chem/WACCM/lidar intercomparison and results.

**Reviewer #1**

**General comments**

**The analysis is thorough and compelling, but I think the authors should also use measurements from the extensive network of ozone monitors in the Los Angeles area to provide more context for their results. This would better link these mountaintop measurements to the issue of regional ozone attainment they invoke in both the introduction and conclusions.**

Thanks for the useful comments. We decided to include supporting information from nearby stations (see also reviewer #2 answers regarding the shift in the ozone maxima, observed at TMF but not at Phelan and Crestline stations) in the surface ozone discussion as well as in the case studies.

**Additional comments**

**P2, L46. ". . .near-surface measurements carried out at. . ."**

Done

**P3, L63. ". . .northwest of Wrightwood..", "The site hosts. . ."**

Done

**P3, L64-67. It is never clearly stated that the Chen study was based on DOAS. Perhaps start with "Differential Optical Absorption Spectroscopy (DOAS) measurements by Chen et al. . ."**

Done

**P3, L67. "Similar conclusions have been reached. . ."**

Done

**P4, Table 1. Is the temporal resolution of the TMTOL really 1 hour or is that just the integration time used for the comparisons?**

1 hour is the integration time. The data acquisition resolution is 3 min, but only very noisy datasets can be obtained with the resolution.

In order to clarify this, "1 hour" was replaced by "1-hour averaging".

**P4, L86 and Figure 2. Is there an explanation for why the TMTOL values are consistently higher than the UAV values below 75 ppb? Did the UAV carry a standard ECC ozonesonde? Also, what is the relevance of showing the "Days since first validation"? Is this meant to somehow account for the one outlier?**

At this time, we have no explanation for this difference, but it is consistent with the agreement observed in the cited paper (Leblanc et al., 2018). The higher values of TMTOL could be related to aerosol extinction effects, but since no systematic study was conducted to determine this, we decided to take it out from the paper.

The idea behind showing the time difference is to provide evidence that the receiver is stable over time. Since the lowest altitudes of the lidar retrieval are very sensitive to misalignment, showing little change over time provides additional confidence regarding the stability of the receiver.

The following clarifications were introduced:

"with an UAV-borne ozonesonde" -> "with an UAV-borne electrochemical concentration cell (ECC) ozonesonde"

"performance" -> "performance over time".

**P7, L164. Would the authors care to comment on the clear seasonal shift between 2012-2014 (highest O3 in May and June) and 2017-2020 (highest O3 in July and August) in Figure 3a?**

The reason for this shift is unclear. The low number of years used here do not allow firm conclusions on the role of interannual variability, which is partly driven by year-to-year changes in meteorology and regional and local wildfire activity. Given the degree of uncertainty associated with this feature, we are not able to comment on it.

**P8, L181 and Figure 4. Are the seasons defined here as Fall (SSO), W(DJF), Sp(MAM), and Summer (JJA) or by equinox/solstice?**

We use the meteorological definition of seasons on the paper. Fall (SON), Winter (DJF), Spring (MAM) and Summer (JJA).

The following clarification was introduced:

*"Here, summer, fall, winter and spring are defined as June-July-August (JJA), September-October-November (SON), December-January-February (DJF), and March-April-May (MAM) respectively."*

**P8, L183 and below. "forecast" is the preferred form of the verb-not "forecasted".**

Done.

**P8, L186. Are the shifts in wind direction consistent with the lower resolution of the model topography?**

Unfortunately, since TMF is surrounded by complex terrain, it is not possible to determine the exact impact of the lower resolution topography without re-running the model at a much higher resolution. While this could be an interesting study, it is out of the scope of this work.

**P8, L195. What was the resolution of the meteorology used for the HYSPLIT back trajectories?**

The HYSPLIT trajectories were calculated based on the WRF-Chem met fields (12x12 km). See Sec. 2.5 for more details.

The following clarification was introduced:

"back-trajectories for summer 2019 and summer 2020" -> "back-trajectories based on WRF-Chem meteorological fields for summer 2019 and summer 2020"

**P8, L206. "The trajectories corresponding to the 210-240° prevailing winds. . ."**

"to the prevailing winds end over this high surface" replaced by "to the prevailing winds (210-240 degrees) end over this high surface"

**P8, L207. ". . .with respect to the 250-300° back trajectories. . ."**

Done

**P9, Figures 3b-3d. These plots are confusing. What statistics are represented by the box-and-whisker plots? I assume the shading refers to 2019 and 2020, but this is not explicitly stated. What do the dark gray diamonds represent? By the way, the plotted symbols are time series and not "scatter plots".**

Shading refers to 2019 and 2020 as in Figure 3a. The box plot provides the quartile distribution of the WACCM stratospheric ozone, temperature and humidity for each month. The single black diamonds represent values out of the 1.5*interquartile range (IQR), also called 'outliers' in this kind of plots. The light and dark orange points are the values that these variables (WACCM stratospheric ozone, humidity and temperature) took during each exceedance day in 2019-2020. The idea behind the plot is to show how far out of the norm the ozone exceedance days are with respect to these variables and indicate whether stratospheric ozone had a role on the exceedances.

In order to clarify these points, we added a legend to panel Figure 3b showing that colors refer to 2019-2020 and added the following sentence to the figure caption:

*Values exceeding 1.5 times the interquartile range (whiskers) are also shown (black diamonds).*

And replaced

''are shown as scatter plots" -> "are shown for comparison (dots)"

**P11, L211. The discussion here skips back from Figures 5 and 6 to Figure 4 without warning. It took me a while to figure out the authors were referring to Figure 4 in the paragraph that followed. Perhaps revise to "During summer 2019, PM10 observations and forecast (third and sixth rows of Figure 4). . .".**

Done

**P11, L213. "A difference is observed for ENE. . ."**

Done

**P11, L229. ". . .focused on the free troposphere. . ."**

Done

**P11, L214. Were there any significant wildfire influences in the Fall of 2019?**

No significative wildfire influences were registered during this period of study.

**P11, L217. The Met One 212 has a lower size cut at 0.3 μm and thus may be excluding the smaller particles in the model analysis.**

While the lower size cut at 0.3 um diameter will certainly have some impact on the PM10 retrieval, the influence is expected to be less than 10% due to the r^3 dependency of the aerosol volume/mass distribution. For example, for a typical volume size distribution derived from AERONET at TMF, only a small fraction of the volume size distribution falls under 0.15 um radius:

[Figure]

The following clarification was added to the manuscript:

*"Since the cut-off diameter of the particle counter is 0.3 um, an underestimation in the derived PM10 values is expected for fine-mode dominated aerosol events"*

**P12, 13. Figures 5 and 6. The gray back trajectories are hard to see in some of the plots. Perhaps use heavier white or magenta lines?**

Lines were change to white (with black borders) to improve visibility.

**P16, L279+. In my opinion, it would be better to switch sections 4.1 and 4.2 and describe the pollution transport event first since this is the more typical event. That would help to put the TMTOL and ceilometer measurements from Figure 9 in better context.**

Done.

**P16, L284. This sentence is awkward and could perhaps be phrased better.**

Done. The sentence was rephrased and simplified.

*"Additionally, and although WRF-Chem and WACCM forecasted enhanced ozone in the lower free troposphere associated with a SI, WRF-Chem did not forecast an effect on the surface ozone as it finally occurred, which illustrates the challenges associated with modeling the entrainment into the nocturnal surface layer and in complex terrain."*

Was changed to:

*"Additionally, this case study illustrates the challenges of forecasting the impact of SIs on surface ozone concentration. In particular, the difficulties associated with an accurate representation of the entrainment into the nocturnal surface layer in complex terrain."*

**P17, L295. The WRF-Chem RH isn't shown.**

Added as vertical profiles for three different times in Fig. 13.

**P18, F9 caption. (a) The CO scale is missing. (b) Please note that the PBL height is from the ceilometer measurements in (c). The ozonesonde profile mentioned in the caption does not appear to be in the plot. (d) The plot is already complicated, but it would be useful to see the WRF-Chem RH since it is mentioned in the text.**

The following changes were introduced: (a) Labels for CO (10,20 and 30 ppb) were added to the plots.

The ozonesonde corresponds to the second case study. This was corrected by changing the ozonesonde phrase from the Figure 9 caption to the Figure 11 caption.

The WRF-Chem RH was added as a separate plot for three different times, together with ozone and potential temperature to illustrate the reasons why WRF-Chem didn't capture an increase in the surface ozone associated with the stratospheric intrusion.

**P19, L340. Again, it would be useful if the TMF surface measurements were compared with the regulatory measurements, particularly those from particularly Phelan and Crestline.**

We added the following to the end of Sec. 4.1:

"*The surrounding Phelan and Crestline stations exhibited an opposite behavior to TMF, with the surface ozone greatly exceeding the EPA threshold and measurements showing generally higher ozone than forecast (Fig. A2).*"

And added a figure to the appendix:

[Figure]

**Figure A2.** Forecast and measured surface ozone concentration at TMF and nearby stations (Phelan and Crestline) for the period comprehended between 27-28 May 2020.

**P21, L357. ". . .the closest time to the ozonesonde. . ."**

Done.

**P23, L281. The high aerosol content of the irruption is not obvious from the ceilometer measurements.**

An indicator ('See text') was added to Fig. 15c.

**P24, L396. This case can be contrasted with that described in Langford et al 2012\* where the descent of a deep SI also caused surface O3 to decrease in the Los Angeles Basin. In that case, however, the surface RH also decreased as drier air from aloft displaced local pollution.**

The following paragraph and appendix figure were added:

*"The ozone monitoring station in central LA (Fig. 3A) did not record any obvious increase in surface ozone associated with any of the two SI events. For the second SI event, the ozone concentration actually dropped more abruptly than forecast, likely related to the onset of the cold front and associated reduced photochemical activity. The Crestline surface monitoring station (the closest high-elevation site to TMF) did show an episodic increase in surface ozone concentrations a few hours before the increase observed at TMF. Surface ozone forecast at the surrounding stations followed the general trend of the observations, but failed to reproduce the SI related increases at the TMF and Crestline station; likely as a result of insufficient entrainment into the nocturnal surface layer. These episodes resemble a case study presented in Langford et al. (2012), where another deep SI was determined to be responsible for an ozone threshold exceedance at the Joshua Tree National Park during 28 May 2010, while the rest of the stations in the LA basin showed a decrease in the ozone concentration as a result of decreased photochemical activity associated with the passage of a cold front."*

[Figure]

**Figure A3**. Forecast and measured surface ozone concentration at TMF and nearby stations (Phelan, Crestline and LA) for the period comprehended between 13-15 June 2020.

**P24, L406. "This result,. . ."**

Done.

**Reviewer #2**

**General comments**

**The manuscript submitted by F. Chouza and co-authors concerns the analysis of ozone measurements at the Table Mountain (TMO) California station. The objectives are multiple (1) to demonstrate the contribution of new measurements at low altitudes of the TMO lidar for the analysis of regional**

pollution (2) to analyze the respective role of stratospheric intrusions and regional pollution on ozone threshold exceedances (3) an evaluation of the performance of WRF-CHEM model ozone forecast for the analysis of pollution episodes.

**The paper contains many interesting results based on a joint analysis of observation and modeling. It certainly deserves a publication in ACP. My main criticism will be on the readability of the objectives and the overall coherence of the paper. The authors are probably trying to meet too many objectives in the same paper. The evaluation of the WRF CHEM forecast seems to be both the most original objective considering the previous publications on the analysis of ozone measurements in the Los Angeles area and a good approach to ensure consistency in the paper. This could be improved by (1) a better presentation of the objectives in the introduction (2) add a discussion of model/measurement differences in section 3.1, otherwise the results presented will be difficult to use (3) Focus section 4 on the evaluation of model performance for the three case studies. The latter could be seen as examples to discuss the differences identified in section 3.2 using the 726 lidar profiles. Indeed the identification of the 3 high surface O3 drivers at TMF based on section 4 is not really new (see previous papers by Cooper, Langford, Lin) and the same results could be very useful if the discussions focus on the model forecast performances to represent theses processes.**

Many thanks for the valuable comments on the general structure of the paper. The introduction was modified as suggested in order to clarify the objectives of this work. Additional comments and comparisons with the models were added to emphasize that aspect of the paper.

**Detailed remarks**

**Line 37 and 51: Relevant objectives are provided here and the introduction generally misses a list of these objectives in addition to the list of tools used.**

The paragraph between L44-52 was modified to include the list of paper objective together with the tools used to address them. The new paragraph is:

*"In this work, surface and lidar measurements conducted at the Jet Propulsion Laboratory Table Mountain Facility (JPL TMF) in the San Gabriel Mountains (Southern California) are used to address three main objectives. Firstly, to demonstrate the new near-range measurement capabilities of the Table Mountain tropospheric ozone lidar and their value for pollution transport and deep stratospheric intrusion studies. Secondly, to investigate the relative impact of regional pollution transport and stratospheric intrusions on the exceedances of the National Ambient Air Quality Standards at TMF, and to determine the representativeness of surface measurements as a proxy for the free troposphere. Finally, to use these surface and vertical profiles to evaluate the performance in complex terrain of the Weather Research and Forecasting model coupled with Chemistry (WRF-Chem) and Whole Atmosphere Community Climate Model (WACCM) forecasts produced daily by the Atmospheric Chemistry Observations and Modeling (ACOM) laboratory at the National Center for Atmospheric Research Atmospheric (NCAR)."*

And slightly modified the following paragraph (paper structure) to emphasize the last two objectives (SI/pollution impact and model performance) :

*"Section 3 presents an analysis of the relative impacts of pollution transport and stratospheric intrusions to the observed ozone threshold exceedances at TMF, as well as an evaluation of the WRF-Chem forecast based on ground and vertical profile measurements conducted between May 2019 and September 2020*

*(the period during which the model data is available). In section 4, three case studies depicting the main mechanisms driving high surface ozone events at JPL TMF are discussed and compared with the WRF-Chem/WACCM forecast of these events."*

**Line 40-44: While the representativity of a mountain top station to characterize PBL or free tropospheric air is a very important point, it is not clear how the paper address this question in section 3.1 or in section 4. If some results address this question they should be mentioned in section 3 or 4.**

Part of this question is addressed in Fig. 3 and the associated discussion. Generally speaking, the large number of ozone threshold exceedances observed at TMF cannot be explained by the influence of stratospheric intrusions and is mainly associated with low level transport from the LA basin area. Considering the strong influence of low-level transport at TMF, it cannot be assumed that the surface measurements at TMF are representative of the free troposphere. In order to provide further evidence of the pollution transport impact, we decided to add a fifth panel to Fig. 3 that shows the WRF-Chem forecasted anthropogenic CO tracer concentrations at TMF as well as the values corresponding to exceedance days.

The following discussion was added towards the end of the subsection:

*"Additionally, the concentrations of the anthropogenic CO tracer as forecasted by WRF-Chem appear to be highly variable and generally non-negligible over the whole year, with median values of 12 ppbv during winter and over 25 ppbv during summer, suggesting that surface and near-ground measurements at TMF are strongly influenced by local sources and cannot be generally assumed to be representative of the free troposphere."*

**Line 175 : Scatter of WACCM stratospheric tracer on O3 exceedance days are a little hard to read in figure 3.b but it seems that the mean is always higher than the median or even than the upper interquartile of the stratospheric tracer for the overall data set. So is it true to say that the stratospheric intrusion plays a limited role based on this unique feature? I am not sure I understand the differences between the black and color dots in Fig. 3. On the other hand it is true to say that the O3 exceedance days with RH>50 % are often observed. So interpretation of the WACCM forecast is not straightforward. May be the age of the stratospheric tracer could help.**

Unfortunately, the WACCM product does not include the stratospheric tracer age. Instead, we decided to add anthropogenic CO tracer values from WRF-Chem as discussed in the previous answer.

**Line 179 : Make a new section here because there is no obvious link between the analysis of the O3 exceedance days shown in Fig. 3 and the new question discussed in Fig. 4 where the wind and diurnal cycle dependency of surface and model are shown.**

A new section was created to separate the pollution/stratospheric influence discussion from the angular dependence discussion. The first subsection was renamed as "*The impact of pollution transport and stratospheric intrusions on high ozone days*", while the second subsection was named "*Surface ozone and PM10 as function of time and wind direction*". Section 3 title was changed from "*WRF-Chem evaluation*" to "*General ozone features and model evaluation*" to account for the discussion beyond the WRF-Chem/WACCM evaluation.

**Line 190 : I agree it is a nice result of this study. However one could expect some hints about of the model difficulties to reproduce this diurnal cycle. Is a well known feature or specific to the TMO data set ?**

The following paragraph was added to the discussion:

*"A comparison of the WRF-Chem surface ozone output with the nearby surface ozone stations of Phelan and Crestline, suggests that this temporal shift in the ozone maximum is a particular feature of the model over TMF, while the underestimation of the ozone levels during morning hours by WRF-Chem is common to all three stations (Fig. A1). The cause of this localized temporal shift is uncertain at this point, but it might be related to the smoothing of the terrain in the model and its impact on the slope mountain effect."*

Additionally, Fig. 1 and Fig. 7 were modified to include Phelan and central LA among the relevant stations for this study:

[Figure]

**Figure A1.** Mean ozone daily cycle at TMF and nearby stations together with the corresponding WRF-Chem output for summer 2019 and summer 2020.

**Line 191 -201: The authors give a great importance to the distribution of ozone in the prevailing wind sector but why do you expect an ozone maxima for this sector ? Indeed according to the ozone distribution shown in figure 5, the ozone max are found west and south of TMO. So the TMO surface ozone max is obviously found for the western and southern wind sectors. I propose to simplify the discussion line 191-201 and use Figure 5 to explain why the observed ozone maxima are for the western and southern sectors. Then the second level of discussion is the comparison of the ozone/wind dependency for the observations and the forecast, which is quite good.**

The ozone maxima were expected to coincide with the prevailing wind direction because it is approximately coincident with the direction of the highest population areas in the LA basin. Certainly because of the complex relationship between NOx and VOCs, the maximum in emissions might not be coincident with the maximum in the ozone production (as became evident when looking at the WRF-Chem plots). Additionally, it is interesting to note that despite being surrounded by complex terrain, the trajectories are fairly straight and mixing does not blur this angular dependence in the ozone arriving to TMF.

Additionally, as supporting validation information on WRF-Chem and following the comments from reviewer #1, we decided to add the actual values from EPA stations to Figs. 5 and 6.

These modifications were accompanied with the following addition:

*"The comparison with the EPA surface ozone monitoring stations (see LA, Santa Clarita, Crestline, and Phelan measurements in Figs. 5a,b and 6a,b) shows a good qualitative agreement with the WRF-Chem output, but with measurements showing generally higher ozone levels at Crestline/Santa Clarita and lower values at the central LA site. The latter can be attributed to enhanced near-surface titration associated with high surface $NO_x$ levels (Figs. 5c,d and 6c,d)."*

**Figure 5 : add TMO position on this figure, trajectories are also hard to read and it is not clear why they are different for top, middle and bottom panel**

The TMF position was added to the figure. Based on this and reviewer #1 comments, we decided to change the color of the trajectories to white (with black borders) to enhance the contrast. Regarding the three rows of panels, the trajectories are the same for all of them. Panels (a), (c) and (e) show the same trajectories (westerly wind), while (b), (d) and (f) show also the same trajectories (prevailing winds).

**Section 3.1 A summary about the main findings about this surface measurement/model comparison must be given at the end of the section: accuracy of model ozone field around TMO, biais in diurnal variation, accuracy of ozone titration in the model.**

The following paragraph was added at the end of the section to summarize the main findings of the measurement/model comparison as well as regarding the influence of pollution transport on the summer surface ozone levels at TMF.

*"WRF-Chem was shown to be able to qualitatively reproduce most of the features observed in the spatio-temporal distributions of ozone and PM10 at TMF and the surrounding stations. Some differences were observed regarding the amplitude of the ozone diurnal cycle at TMF and the nearby stations (Figs. 4 and S1). In the particular case of TMF, the forecasted maximum of the ozone diurnal cycle was about 3 hours earlier than the measured maximum. As also shown in Sec. 3.1, the results shown in this subsection indicate that surface ozone concentrations during summer at TMF are strongly influenced by pollution transport from the LA basin region. Transport from central LA, where titration limits the surface ozone concentrations, is characterized by generally lower ozone levels at TMF, while transport from Santa Clarita is generally associated with higher ozone concentrations."*

**Section 3.2 It is a very interesting section and there is little published work where many lidar profiles are statistically compared with regional-scale CTM simulations. However the discussion is not very developed and the lidar vertical resolution is probably not the main driving factor. While large**

**differences due to small time or spatial shift of the simulated ozone layers above TMO are expected near the tropopause or at the PBL top, it is unexpected to find 30-50 % differences at 7.5 km. A small discussion about possible reason for this overestimate of the forecast could be added. Was this feature already observed during model evaluation in the free troposphere ? Is it specific to TMO and why ? The differences seems also worse during nighttime (except in summer 2020) while the lidar accuracy is less for daytime observations. Do you have an explanation for this ?**

The WRF-Chem runs are initialized using WACCM. We ran a comparison of the lidar measurements with WACCM and we observed the same difference, suggesting that the problem comes from WACCM. The following was added to the section:

*"The observed variability in the free troposphere also shows a clear seasonal dependence, with larger variability during summer and reduced variability during winter. The same seasonal pattern is also visible in the WRF-Chem profiles.*

*The WACCM forecast shows a very similar behavior to the one described for WRF-Chem, including the altitude-dependent bias. Since the WRF-Chem chemical boundary conditions are determined by the WACCM forecast (Sec. 2.5), the bias observed in the WRF-Chem runs is likely a result of the bias in the WACCM forecast."*

[Figure]

***Figure 8.*** *Relative difference (third column) between TMTOL (first column), WRF-Chem (red) and WACCM (green) ozone profiles (second column) for the period between summer 2019 and summer 2020 (rows). TMTOL retrieval vertical resolution as well as the vertical grid of the models are also shown for each season*

*(fourth column). Actual ground level and WRF-Chem surface level are shown as grey and dark grey shaded areas respectively. 1-sigma variability on the ozone profiles and vertical resolution of TMTOL is indicated by the shaded areas.*

Additionally, we also performed a comparison of WACCM with the ozonesondes launched at Trinidad Head, CA and Boulder, CO. The preliminary results indicate a similar bias (see below), suggesting that this bias is not particular to TMF, but a more general model feature:

The following was added to the section:

*"In order to investigate if this ozone excess is a particular feature of WACCM over TMF, we performed a comparison of the WACCM ozone forecasts and the ECC ozonesondes launched regularly at Trinidad Head, California (about 1000 km north-west from TMF) and Boulder, Colorado (about 1200 km north-east from TMF) for May 2019 to August 2020. The results (Fig. 9) indicate a similar altitude-increasing bias, suggesting a synoptic scale deviation of the forecast for the period under study as a possible reason."*

[Figure]

***Figure 9.*** *Relative mean difference between WACCM and ECC ozonesondes during the period comprehended between May 2019 and August 2020 over Boulder, Colorado and Trinidad Head, California. The 1-sigma standard deviation of the difference is indicated by the shaded area.*

**Section 4.1. It is indeed an interesting case study to demonstrate the role of stratospheric intrusion for increasing surface ozone measurements at a mountain top station. However it is not a new result which has been reported in several publications. So a discussion is missing either to compare this new case study with previous ones (Bonasoni 2000, Trickl 2019, Knowland 2017) or to explain why the model did not forecast a surface ozone enhancement (line 285 and 300).**

The following discussion was added together with an additional figure (note that the figure numbers changed as the case study 1 and case study 2 were switch as suggested by reviewer #1).

*"This difference is better depicted in Fig. 13, where ozone, relative humidity and potential temperature as forecasted by WRF-Chem are presented for three different times. The first profile corresponds to the pollution transport event during the late afternoon (A, 0 UTC), with a 500 m deep PBL characterized by an ozone concentration of over 70 ppbv, a relative humidity of almost 30% and a moderately strong temperature inversion at its top. Just above the PBL, and characterized by a relative humidity of 10%, we can see the SI influence forecasted by WACCM and shown in Fig. 12a. As the PBL collapses and the SI approaches the surface, a strong temperature inversion develops near the ground, which inhibits mixing of ozone from the SI and limits its impact in the surface."*

[Figure]

***Figure 14.*** *WRF-Chem ozone, relative humidity and potential temperature over TMF at times A (green, 0 UTC), B (blue, 8 UTC) and C (red, 12 UTC) indicated by red arrows in Fig. 13a top. The actual TMF elevation (light grey shaded) is shown together with the model elevation (grey shaded).*

**Line 308 Can the authors use their detailed spatio-temporal analysis of the measured and forecasted ozone field in Figures 9 and 10 in order to discuss the differences in the time series at TMO ? Can a model shift along BB' explain the observations ? Can such a shift explain why the model is too low at the surface or is the model mixing too low in the PBL ?**

The cause of the low ozone at the surface is explained in the previous answer.

**Figure 10c The dashed line above mountain top is not defined. Is it the PBL top like in figure 9 ? Add a vertical line in Figure 9 at the time corresponding to the horizontal/vertical cross-sections shown in Fig.10. Similar comment for fig. 13 and 11.**

Yes, the dashed lines indicate the PBL by the model. The definition was added to the caption. The time of the cross-sections was also added (black arrows on top).

**Section 4.2. This case is indeed interesting for the model assessment because the large scale upper tropospheric ozone feature are weaker than case 4.1 and small shift of the modeled ozone field will not easily explain the lidar/model difference.**

Section 4.2 (now Section 4.1) is indeed an interesting case, although we consider that the agreement of the upper troposphere feature in this case study as fairly good considering the results presented in Fig. 8. In addition it needs to be considered that this is a forecast product. See also the comparison to the ozonesonde measurements. The main differences are localized in the PBL and lower troposphere, where we have the temporal shift in the ozone maxima as well as a low-level inversion that inhibits mixing and that is challenging to capture by coarser resolution models.

**Line 339. Does this kind of model overestimate correspond to the statistical positive bias between model and TMTOL near 3 km which is discussed in section 3.2 ?**

This is difficult to say, because most of the profiles of section 3.2 (now section 3.3) corresponds to either noontime (20 UTC) or after sunset (5 UTC +1 day for this case study), which corresponds to either before or after the maxima of pollution transport. For a comparison in the free troposphere, this time difference is not expected to produce a major impact. In the PBL, below 3 km, the impact of this time difference is expected to be larger due to the larger temporal variability.

**Line 341 It is worth mentioning that the forecast daily maximum is earlier in the model as previously shown in section 3.1 using the surface observations. It seems to be a permanent feature of the forecast.**

Yes, a reference to Sec. 3.1 (now Sec. 3.2) was added to the paragraph.

**Line 378 What is the reason for the model performing less for this case study ?**

The following was added to the case study discussion:

*"This underestimation of the forecasted ozone is likely related to a difference between the forecasted and the actual wind fields. The forecasts show a fairly constant wind direction of 210 degrees (transport from central LA) after 11 June 22:00 UTC, while the measured wind direction at TMF was about 180 degrees (transport from the Fontana/San Bernardino area, south of Crestline). This difference corresponds to two different transport regimes as can be seen in Fig. 4."*

**Line 344 Is mixing too high in the model close to the ground ?**

There is no evidence of a systematic problem regarding the near-ground mixing. As can be seen from the first and second case studies, in one case the model forecasted too much mixing, while in the other it forecasted too little.

**Line 413 Although the model/TMTOL differences are lower than 20% within the PBL, it must be stressed that there is a model positive bias which might be related to the large differences observed in the free troposphere between 4 and 7 km.**

Some changes were introduced to the conclusion section, including a discussion on the influence of the free troposphere bias in the PBL.

**Reviewer #3**

**General comments**

The submitted manuscript submitted by F. Chouza and co-authors concerns the analysis of ozone measurements at the TOLNet/NDACC JPL/Table Mountain (TMO) California station.

The manuscript focuses on the impact of Los Angeles basin pollution transport and stratospheric intrusions on the surface ozone levels observed in the San Gabriel Mountains is investigated based on a combination of surface and lidar measurements as well as WRF-Chem (Weather Research and Forecasting with Chemistry) and WACCM (Whole Atmosphere Community Climate Model) model runs.

This manuscript poses a variety of central thematically relevant questions that could be explored further. The introduction hints at but does not investigate increasing the accuracy of detangling the high-altitude sites (generally use for long term background/clean trends) and their respective local ozone episodes. There is comparisons with model data, but there really isn't a way to extract conclusions that are satisfying in my opinion. The manuscript also begins to describe the ozone NAAQS and ozone exceedance episodes from a regulatory perspective, but does not really go beyond a simple description and table which is not referenced in the case studies later in the manuscript. Many of this has been done previously by the papers referenced within the document (e.g. Cooper, Lin, Langford).

I suggest the authors reorder the manuscript to re-emphasize the new case studies and comment on their regulatory implications in a discussion section. What is clear is the rich and novel data set in the latter half of the manuscript, especially the 2020 episode. This is very appropriate for this journal with a description of new science learned from this case study. This shows clear evidence of the upslope flow/LA basin impact followed directly by a stratospheric intrusion.

Thank you for the comments on the general structure and content of the paper. Based on this comment and the comments from the reviewer #2, we decided to reformulate the introduction to better communicate the main objectives of the paper. In this case, the objectives of the paper are multiple: (1) illustrate the capabilities of the new very-near range receivers to address near-surface pollution transport events. (2) Determine how much of the exceedances relate to SI vs local pollution transport. (3) Evaluate WRF-Chem/WACCM from a general perspective and for the main mechanisms controlling the surface ozone at TMF.

While the case studies are an important part of the paper, they only address part of these points, so we decided to leave them as a second section that complements the first section (more general results).

**L23 – Carbon monoxide is not central to the photolysis to generate ozone. CO is certainly a useful tracer. Did you mean OH radical?**

'*carbon monoxide*' was removed from the sentence.

**L25 – for this site location, is transport notionally from the LA Basin in a regional sense or inter-continental transport on a synoptic scale?**

For TMF, as discussed through the paper, the main driver for near-ground ozone variability has been identified as regional transport from the LA basin area. Nevertheless, the impact from the LA basin can extend as far as Las Vegas and probably further away. Unfortunately, the further away the site under study is, the harder is to pinpoint the source to a specific region.

**L45 – a "few", is there a way to more quantitatively define this?**

The introduction was modified according to your general comments and the comments from the Reviewer #2. There is no more reference to a 'few' stratospheric intrusions.

**L75 – is this lidar also part of the NASA TOLNet network? I assume so considering the work in LeBlanc et. Al. 2018.**

**Yes, the following was added to the end of the sentence:**

*"and later included to the Tropospheric Ozone Lidar Network (TOLNet)."*

**Figure 2: What is trying to be conveyed in this z axis? That there is negligible instrument drift? Would a time series be more appropriate? Are the 90ppbv measurements taken during the lidar data shown in subsequent sections? Having a 100m to surface comparison would also be useful to illustrate the gradient on select days.**

The idea behind this plot is to show that there is little instrument drift in the bottom part of the retrieval. Since the validation was conducted mainly in clusters of several measurements spaced by a considerable amount of time, a time series would not be the most appropriate way to present them. These measurements do not correspond to any of the case studies shown in Sec. 4. Unfortunately, the validation using the UAV requires physical presence on the measurement site, and all the case studies were conducted remotely due to COVID-associated lockdowns.

Some of this gradient evaluation is part of section 4, where surface measurements are shown together with the 100 m AGL lidar retrieval. Indeed, some interesting gradients were observed (Sec. 4.3 for example).

**L118 – If the model does not include stratospheric chemistry, is this really an appropriately rigorous evaluation?**

WACCM does include stratospheric and tropospheric chemistry. WRF-Chem boundary conditions are initialized using WACCM forecasts every day. Since the use of WRF-Chem in this study focuses in the lower troposphere, and all stratosphere-related evaluations are based on the WACCM model, we don't see any particular issues with the approach presented on the paper. Actually, we added WACCM to the comparison presented in Sec. 3.3 and the results of WRF-Chem in the upper troposphere are very similar to the results derived for WACCM (which is expected considering that WRF is initialized with WACCM).

**Figure 5 – The back-trajectories are difficult to see. Consider alternative approach. Since most of the trajectories are in the same general cardinal direction, why not use a vertical image instead to illustrate the particle dispersion and upslope/mountain chimney effects?**

A change in the color of the trajectories was made to improve visibility.

**Figure 8 – It is challenging to see a clear result from this figure. Is this an emphasis on showing seasonal changes in ozone or an evaluation of the model? Is the model just not brining enough stratospheric ozone into the troposphere? There is a systematic bias above 7500km. Could this figure be replicated but for a time series of Tropospheric ozone Columns? This product may also be relevant to the upcoming geostationary satellites**

Thanks for pointing this out. The figure was mainly focused on the evaluation of WRF-Chem. The idea behind dividing the plot by season was to see if the bias was dependent on the season (and time of the day). We decided to modify this figure to emphasize the validation aspect. This modification includes adding WACCM to the comparison (since it is used to initialize WRF) and remove the day/night profiles (little change was observed between day and night).

**Figure 14 – This is the clearest evidence of a LA Basin pollution transport followed by a stratospheric intrusion reaching the western US I have seen. Very appropriate for the journal and the community needs to be aware of these measurements. This should be re-emphasized as the central part of this manuscript.**

Thank you for your comments on this. While this is certainly a good example of LA basin pollution/SI influence on near-ground ozone and the model performance, it only addresses part of the objectives listed in the introduction. For this reason, we decided to leave it as part of Sec. 4.

[revised manuscript text omitted]

---

## Referee Report (RR1)

Review of ACP-2020-1208 revised

The authors have done a good job of addressing the reviewer comments, which has significantly improved the readability of the paper. I would recommend only a few small changes to the revised text.

P1, L35. "refereed" should be "referred"

P3, L74. The definition of DOAS is repeated.

P3, L83-84. "…included *in* the…"

P8, L183. This SI seasonality has also been noted in a large number of climatological studies (e.g. Sprenger and Wernli, 2003).

P8, L184. "forecast"

P8, L185 and P9, L215. Perhaps use "upslope flow" instead of "the slope mountain effect"

---

## Author Response (AR2)

Dear editor and reviewers,

We would like to thank you for your comments on the paper. The following changes where included to the final version of the manuscript:

**P1, L35. "refereed" should be "referred"**

Done.

**P3, L74. The definition of DOAS is repeated**

The repeated definition was removed.

**P3, L83-84. "…included in the…"**

'To' was changed to 'in' as suggested.

**P8, L183. This SI seasonality has also been noted in a large number of climatological studies (e.g. Sprenger and Wernli, 2003).**

An additional citation to Sprenger and Wernli (2003) was added.

**P8, L184. "forecast"**

'Forecasted' was changed to 'forecast'.

**P8, L185 and P9, L215. Perhaps use "upslope flow" instead of "the slope mountain effect"**

'the slope mountain effect' was replaced with 'upslope flow'.